# Increased excitatory to inhibitory synaptic ratio in parietal cortex samples from individuals with Alzheimer's disease

Julie C. Lauterborn [1✉], Pietro Scaduto[2], Conor D. Cox[1], Anton Schulmann [3], Gary Lynch[1,4], Christine M. Gall [1], C. Dirk Keene [5] & Agenor Limon [2✉]

Synaptic disturbances in excitatory to inhibitory (E/I) balance in forebrain circuits are thought to contribute to the progression of Alzheimer's disease (AD) and dementia, although direct evidence for such imbalance in humans is lacking. We assessed anatomical and electrophysiological synaptic E/I ratios in post-mortem parietal cortex samples from middle-aged individuals with AD (early-onset) or Down syndrome (DS) by fluorescence deconvolution tomography and microtransplantation of synaptic membranes. Both approaches revealed significantly elevated E/I ratios for AD, but not DS, versus controls. Gene expression studies in an independent AD cohort also demonstrated elevated E/I ratios in individuals with AD as compared to controls. These findings provide evidence of a marked pro-excitatory perturbation of synaptic E/I balance in AD parietal cortex, a region within the default mode network that is overly active in the disorder, and support the hypothesis that E/I imbalances disrupt cognition-related shifts in cortical activity which contribute to the intellectual decline in AD.

[1] Department of Anatomy and Neurobiology, University of California at Irvine, Irvine, CA, USA. [2] Department of Neurology, Mitchell Center for Neurodegenerative Diseases. School of Medicine, University of Texas Medical Branch at Galveston, Galveston, USA. [3] National Institute of Mental Health, Human Genetics Branch, Bethesda, MD, USA. [4] Department of Psychiatry & Human Behavior, University of California at Irvine, Irvine, CA 92697, USA. [5] Department of Laboratory Medicine and Pathology, University of Washington School of Medicine, Seattle, WA, USA. ✉email: jclauter@uci.edu; aglimonr@utmb.edu

Alzheimer's disease (AD) is the main cause of cognitive impairment and dementia in the elderly[1]. In addition to the neuropathologic hallmarks of AD, amyloid β (Aβ) plaques and neurofibrillary tangles (NFT), affected brain regions demonstrate severe synapse and spine density loss[2]. These synaptic changes are major correlates of progressive cognitive dysfunction and dementia in the disorder[2,3]. Clinical studies and experiments with animal models suggest that synaptic loss may disturb the excitatory to inhibitory balance (E/I ratio) in circuits vulnerable to AD pathology[4–6], which in turn could lead to the cortical hyperexcitability that is associated with cognitive impairment[4,7,8]. However, electrophysiological evidence from animal model studies indicates that the activity and strength of excitatory and inhibitory synapses in the cerebral cortex are highly correlated across different cortical activity patterns[9,10]. Thus, synaptic currents, through excitatory AMPA receptors (AMPARs) and inhibitory GABA$_A$ receptors (GABA$_A$Rs), are tightly regulated to preserve the global synaptic E/I ratio within a range that allows for normal network level operations[9,11]. Our previous work showed the total activity of functional GABA$_A$Rs, both synaptic and extrasynaptic, is significantly reduced in the temporal cortex of the AD brain, and that this reduction is similar in magnitude to the reduction in AMPARs[12]. However, those experiments did not distinguish between synaptic and extra-synaptic GABA$_A$Rs, and thus do not directly address potential changes in the strength of fast inhibitory synaptic currents. Moreover, gene expression for the GABA$_A$R subunit *GABRG*2 was more affected than that of AMPAR subunit *GRIA*2[12], a result that raises the possibility of AD related shifts in the operating characteristics of synaptic receptors. In all, the issue of whether AD differentially affects inhibitory vs. excitatory synapses, and thus E/I balance in cortical areas, has yet to be resolved. Such information is critical for refining hypotheses about the origins of AD-related pathogenesis and pathophysiology, and for development of effective therapies.

In the present study, we utilized two complementary approaches to assess the global synaptic E/I ratio in the inferior parietal cortex (PCx) of AD subjects. The PCx is part of the default mode network (DMN) that is active during daydreaming, mind wandering, and introspection, but deactivates during demanding goal-directed cognitive tasks[13,14]. Importantly, baseline DMN activity is increased in AD and fails to deactivate during cognitive tasks, suggesting that the DMN is abnormally and continuously hyperactive in AD[15,16]. This may be due to disruptions in the E/I balance of principal neurons, although direct evidence is lacking. The present study was therefore undertaken to determine the anatomical E/I ratio in postmortem PCx from middle-aged subjects with AD (early-onset), Down Syndrome (DS), and normal controls without pathology, using fluorescence deconvolution tomography (FDT) to histologically assess the synaptic levels of two scaffolding proteins implicated in regulating synaptic E/I balance, the excitatory postsynaptic density protein 95 (PSD-95) and inhibitory postsynaptic protein gephyrin (GPHN) (see Yu and Delbas[17]). The DS group was included to compare synaptic effects in two neurological disorders with AD neuropathologic change (ADNC); virtually all DS individuals exhibit ADNC pathology by 40 years of age[18]. The electrophysiological E/I ratio was then measured by using microtransplantation of synaptic membranes (MSM) isolated from slices adjacent to those used in FDT experiments. As originally described by Miledi and colleagues[19,20], and expanded by others[21,22], the MSM technique allows for assessment of human receptors that are still associated with their natural lipid environment and accessory proteins. Results from the present work provide evidence that despite a loss of both excitatory and inhibitory synaptic proteins, individuals with AD exhibit a marked shift toward a pro-excitatory perturbation of postsynaptic densities and electrophysiological synaptic E/I balance in the PCx. Further corroborating evidence for this imbalance in AD parietal cortex was found using publicly available in situ hybridization and RNA-Seq transcriptional datasets.

## Results

**Anatomical alterations in excitatory and inhibitory synaptic markers in cortical layers of individuals with AD pathology.** Quantitative FDT analyses were used to assess postsynaptic levels of the excitatory marker PSD-95 and inhibitory marker GPHN in PCx from individuals with AD dementia or DS with ADNC, and from non-demented controls (Fig. 1; see Table 1 and Supplementary Data 1 for summarized and detailed demographic information). All AD and DS cases exhibited numerous amyloid plaques in PCx with no significant differences between these groups ($P = 0.800$), whereas plaques were absent in controls (Fig. 1b); thus, the controls were considered subjects without ADNC.

FDT analyses of cortical layers 1 and 2 were assessed individually to test for differences in levels of the synaptic markers between the cell body-sparse and the adjacent cell body-dense layer, respectively. Counts of PSD-95 and GPHN immunoreactive puncta in the sample fields from both layers were not significantly different across groups (Fig. 1c, d; Table 2), indicating that the total density (puncta per volume) of excitatory and inhibitory synapses was maintained in PCx of middle-aged individuals including those in the disease states. Nevertheless, in all cases, there were large lipofuscin deposits often in association with GPHN immunopositive (+) cell bodies, and in the AD cases we observed dystrophic GPHN+ processes.

Next, we assessed the levels of immunoreactivity (ir) for both synaptic markers within individual synapse-sized puncta and plotted these measures in intensity frequency distributions (Fig. 1e, h). Both AD and DS cases were characterized by a reduction in the proportion of excitatory synapses with high levels of PSD-95-ir (≥90 immunofluorescence intensity, on a 20-180 scale) and an increase in the proportion of excitatory synapses with lower intensity labeling. These changes were larger in the perikarya-rich layer 2 as compared to layer 1 (Fig. 1e, g); RM-ANOVA, $P < 0.0001$ for interaction between groups and intensities for both GPHN and PSD-95 in layer 2, vs $P = 0.0066$ for GPHN and $P = 0.0419$ for PSD-95 in layer 1. Consequently, the ratio of the high-to-low immunoreactivity for excitatory PSDs was significantly reduced in all AD and DS cases (Fig. 2a–d). The leftward shift in intensity suggests that a larger proportion of excitatory synapses in AD and DS have smaller postsynaptic densities or disturbances in synaptic scaffolding. The intensity frequency distribution for analysis of GPHN-ir at inhibitory synapses demonstrated that changes in AD and DS cases were similar to those of PSD-95-ir (Fig. 1f, h), with there being a marked reduction in the ratio of high-to-low levels of GPHN-ir per synapse in both layers 1 and 2, and stronger effects in layer 2 (Fig. 2e–h). The disturbances in both PSD-95 and GPHN immunoreactivities suggest reductions in both excitatory and inhibitory drive in AD and DS, with effects being relatively greater in the vicinity of perikarya.

Given the marked reductions in levels of synaptic PSD-95-ir and GPHN-ir in both AD and DS groups, we next asked whether these proteins were similarly reduced or if they were differentially affected on a subject-by-subject basis. The ratio of the peaks of the immunolabeling frequency distributions for PSD-95 to GPHN (PSD-95-ir/GPHN-ir), for each subject, was not different between groups for layer 1 (Welch ANOVA allowing unequal variances, $P = 0.22$), indicating a similar anatomical E/I ratio for

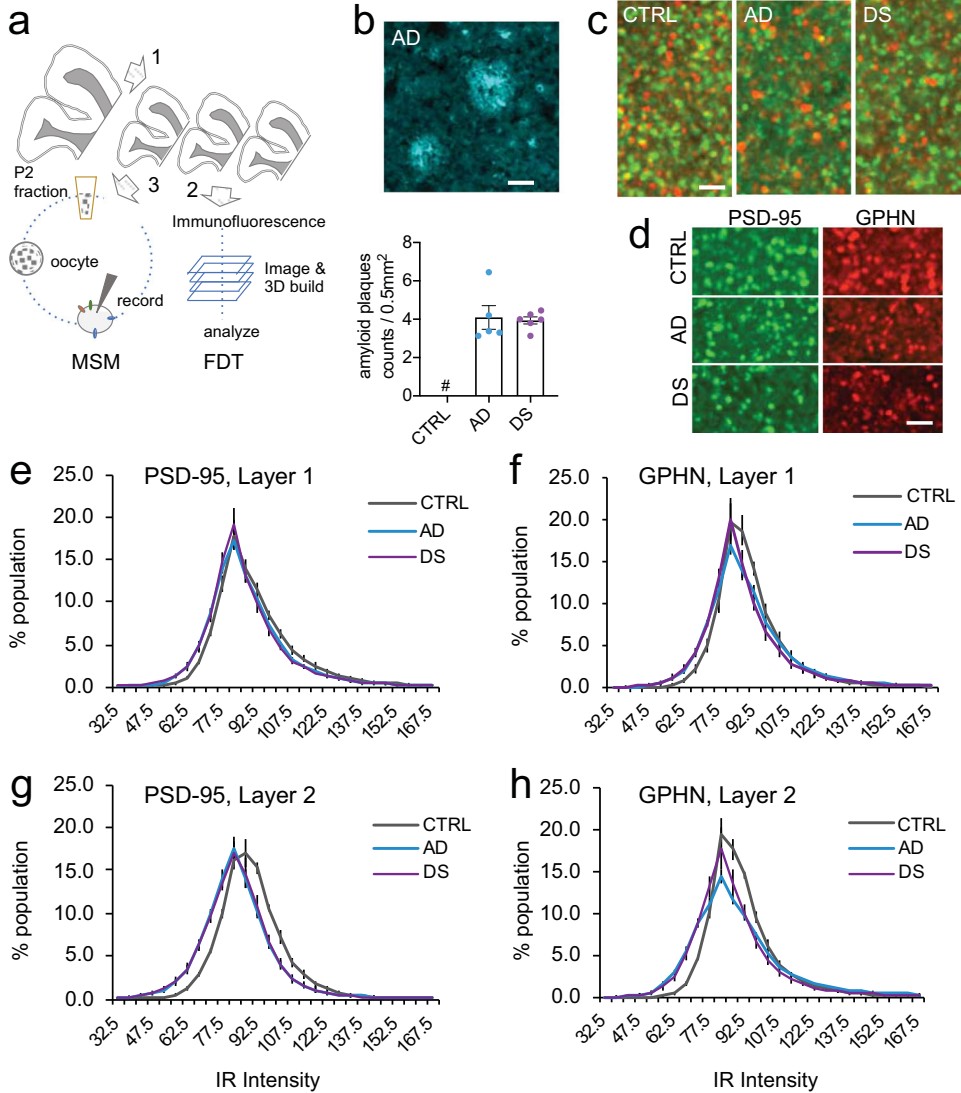

**Fig. 1 Approach for analyses and quantification of changes in synaptic protein levels in AD and DS. a** Schematic illustrating the methodological approach for processing fresh-frozen parietal cortex subsections from the same tissue block (step 1) for either fluorescence deconvolution tomography (FDT; step 2) or microtransplantation of synaptic membranes (MSM; step 3) analyses. See "Methods" for details. P2, synaptosomal fraction. **b** Top: Representative pseudocolor image of amyloid plaques in parietal cortex of an Alzheimer's disease (AD) case (calibration bar, 35 μm). Bottom: Bar graph showing density of amyloid plaques in the three groups (mean ± SEM; quantified from 20 individual 574,200 μm² images per case). Each dot represents the density of amyloid plaques in each subject ($n = 5$ controls, 5 AD and 6 DS). CTRL, control; DS, Down syndrome. Controls had no plaques (indicated by #). All AD and DS cases exhibited plaques, but there were no group differences (AD vs DS, $P = 0.800$, two-sided Student's $t$-test). Photomicrographs showing PSD-95 (green) and gephyrin (GPHN, red) immunoreactive puncta in layers 1 (**c**) and 2 (**d**) parietal cortex from control (CTRL), AD, and DS subjects. Calibration bars, 2 μm. Synaptic immunolabeling intensity frequency distributions from the FDT analyses show the proportion of PSDs (Y-axis) in layers 1 (**e,f**) and 2 (**g,h**) that were immunolabeled for PSD-95 or GPHN at different immunoreactivity (IR) intensities (X-axis) for the three groups; data are expressed as percentages of total labeled PSDs and plotted as group mean values ± SEM ($n = 5$ controls, 5 AD, and 6 DS). Layer 1 results are based on 10 individual 42,840 μm³ 3-D builds per subject with the following exceptions: 12 builds (two AD cases), 11 builds (one AD, two DS cases), and 9 builds (one DS case). Layer 2 results are based on 10 individual 3-D builds per subject, except for one AD case with 12 builds. Note the leftward skew in the immunolabeling intensity frequency distributions for the AD and DS groups relative to controls; this was most pronounced for layer 2 (two-way RM-ANOVA, $P < 0.0001$ for interaction between groups and intensities for both GPHN and PSD-95 in layer 2; $P = 0.0066$ for GPHN and $P = 0.0419$ for PSD-95 in layer 1). Source data for this and all figures are found in the source data file.

AD and DS versus controls (Fig. 3a). However, within layer 2 this ratio was significantly larger in the AD group as compared to controls (Welch ANOVA, $P = 0.0316$; followed by Dunn post hoc test comparing AD and DS vs control); the DS group had a trend in the same direction for layer 2 but this effect did not reach statistical significance due to greater group spread (Fig. 3b). These results suggest that while both PSD-95 and GPHN levels are reduced in AD, in layer 2 of PCx the ratio of excitatory to

inhibitory postsynaptic elements is significantly elevated in AD versus controls.

**Reduction of synaptosome-like particles in both AD and DS.** To determine whether the anatomical E/I synaptic marker protein alterations are indicative of changes in the abundance of functional synaptic AMPARs and GABA$_A$Rs, P2 fractions

**Table 1 Summary of demographics, pathology, and cognitive skills for study groups.**

| Diagnosis | n | Age | Sex | PMI (h) | ADNC | mBADLS |
|---|---|---|---|---|---|---|
| Control | 5 | 57.2 ± 4.9 | 4 M, 1 F | 15.4 ± 4* | | |
| AD | 5 | 57.8 ± 2.2 | 2 M, 3 F | 3.8 ± 1 | Stages VI, C | 27.4 ± 13.7 |
| DS | 6 | 56.3 ± 3.5 | 3 M, 3 F | 4.1 ± 1 | Stages VI, C | 33.2 ± 9.3 |

Values for age, postmortem interval (PMI), and the modified version of the Bristol Activities of Daily Living Scale (mBADLS) are mean ± SD; individual values are presented in Supplementary Data 1. AD Neuropathologic Change (ADNC), Braak stage VI and CERAD stage C. PMIs were longer for the control group (*$P < 0.05$ versus AD and DS groups, Dunn's Multiple Comparison test), but within a range that does not affect synaptic measures; See "Methods" for further details.

**Table 2 Counts of excitatory and inhibitory synapses in superficial layers of parietal cortex.**

| Protein | Region | Control | AD | DS | 1-way ANOVA |
|---|---|---|---|---|---|
| PSD-95 | layer 1 | 24759 ± 534 | 24614 ± 577 | 26856 ± 1177 | $P = 0.1581$, n.s. |
| PSD-95 | layer 2 | 24455 ± 790 | 25012 ± 635 | 24863 ± 572 | $P = 0.8382$, n.s. |
| GPHN | layer 1 | 27037 ± 590 | 25023 ± 1146 | 26353 ± 1065 | $P = 0.3885$, n.s. |
| GPHN | layer 2 | 27148 ± 779 | 24660 ± 298 | 25224 ± 1361 | $P = 0.8499$, n.s. |

Average counts (mean ± SEM) of immunoreactive synapses and statistical comparisons are shown. (n.s., not significant).

enriched in synaptosomes were isolated from a single (20 μm-thick) cryostat section, adjacent to those used in FDT analysis, with the goal of recording the electrophysiological activity of synaptic receptors by MSM. As an intermediate step between FDT and MSM, one aliquot of the P2 fraction was examined by flow cytometry to aid in the interpretation of the immunohistochemical analysis from layers 1 and 2, and the global electrophysiological recording of synapses from the whole slice. Flow cytometry analysis showed a striking 40% and 36% reduction of size-gated synaptosome-like particles in the AD and DS groups (Supplementary Fig. S2), respectively, as compared to the control group ($F$ (2,13) = 12.04, $P = 0.0011$; Fig. 4a, b). Aligning with the loss of synaptosome particles, total protein levels in the P2 fractions were also significantly reduced in the AD and DS groups ($F$ (2,13) = 21.5, $P < 0.0001$) (Supplementary Fig. S1). The amount of protein and the number of synaptosomes were positively correlated $R^2$ (16) = 0.68; $p < 0.0001$.

These results indicate that preservation of the density of PSD-95+ and GPHN + synapses in AD and DS, at least in superficial cortical layers, is at the cost of tissue shrinkage. We also observed a leftward shift in the size of recovered synaptosome-like particles from the AD and DS groups indicating that a large proportion of resilient synapses across all cortical layers have smaller-than-control sizes, a finding that is consistent with the reductions in immunoreactivity for both synaptic markers. Further in agreement with the FDT analyses, the ratio of large-to-small synaptosome-like particles was strongly reduced in AD and DS ($F$ (2,13) = 29.83, $P < 0.0001$ vs control) (Fig. 4f), and the large/small ratio from the whole slice was linearly correlated to the high/low immunoreactivity for PSD-95 and GPHN in layers 1 and 2 combined ($R^2$(16) = 0.72, $P < 0.0001$) (Fig. 4g). Results from flow cytometry strongly suggests that synaptic deficits observed in cell-dense layer 2 identified with FDT are representative of effects in all cell-dense cortical layers, and that the P2 fractions capture those changes.

**Increased electrophysiological E/I ratio in AD but not DS.** Flow cytometry analysis identified a large reduction of synaptosome-particles in P2 fractions in AD and DS; this is in agreement with neuronal loss and synaptic dysfunction found in previous studies. To determine whether changes in postsynaptic markers are associated with changes in AMPARs and GABA$_A$Rs, we

microtransplanted the same amount of synaptosomal membranes for each subject and measured the agonist-elicited responses of excitatory and inhibitory receptors in *Xenopus* oocytes. Because it is known that naïve oocytes (non-injected) do not express endogenous AMPARs or GABA$_A$Rs[23], the ion currents elicited by specific agonists for these receptors in microtransplanted oocytes are mediated by human receptors. To confirm this point, in each experiment we tested agonists for AMPARs or GABA$_A$Rs and were unable to elicit currents from non-injected oocytes, as previously reported. In contrast, oocytes microtransplanted with synaptosomes from each of the three groups, and clamped at a voltage of −80 mV, exhibited fast activated currents when perfused with 1 mM GABA (GABA currents) or 100 μM kainate (kainate currents) (Fig. 5a). Kainate is an agonist of AMPARs that keeps the channel in a non-desensitized state, allowing steady-state measurement of AMPARs currents[24]. Co-perfusion of s-AMPA plus cyclothiazide (CTZ), after 3 min CTZ preincubation, elicited ion currents with similar amplitude of those produced by kainate alone (kainate *vs* AMPA + CTZ correlation: $R^2$(16) = 0.943, $P < 0.0001$; Fig. 5d) suggesting that kainate currents are mostly generated by AMPARs. To determine the contribution of kainate-type receptors specifically, we incubated micro-transplanted oocytes in concanavalin A (ConA), a positive allosteric modulator of kainate-type receptors[25]. Figure 5c shows no effect of ConA in this preparation, confirming that the contribution of kainate receptors in our recordings was negligible.

Having demonstrated agonist-induced responses in micro-transplanted oocytes, we then tested for group differences. Maximal responses to GABA, kainate and AMPA + CTZ were variable across subjects (Fig. 5); this was most striking in the DS group, which included some of the largest ion currents recorded. Therefore, although both AMPAR and GABA$_A$R responses tended to be smaller in AD versus control cases, the differences were not statistically significant. It is important to note that this study was not powered to detect differences in the total amplitude of the currents when measured individually (Fig. 5b). However, the maximal amplitude of currents through AMPARs and GABA$_A$Rs were highly correlated across subjects (Fig. 5e). Because the correlation between ion currents elicited by kainate and GABA ($R^2$(16)=0.93, $P < 0.0001$) was higher than that between s-AMPA + CTZ and GABA ($R^2$(16) = 0.92, $P < 0.0001$), we used kainate and GABA currents to calculate the

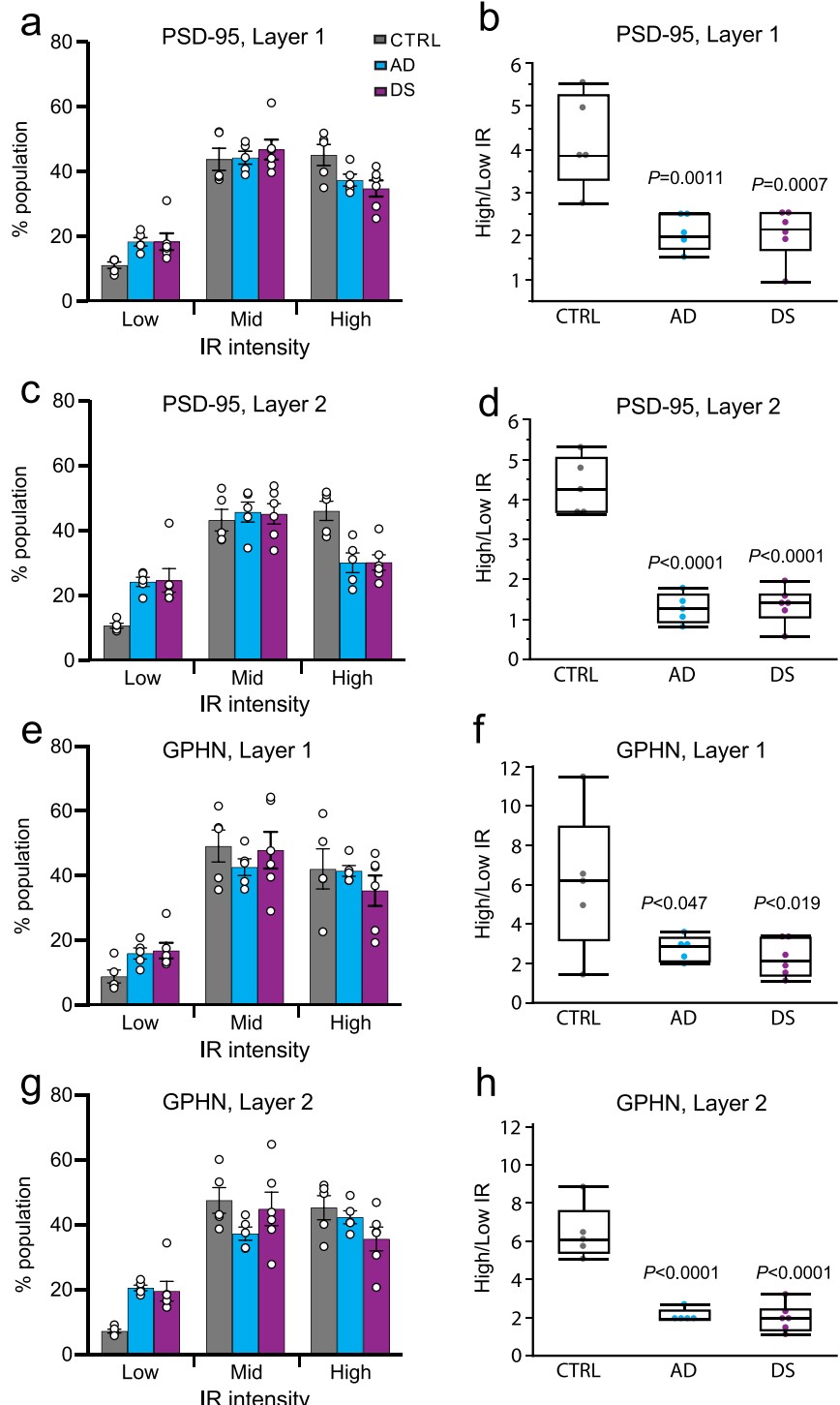

**Fig. 2 Levels of immunoreactivity for excitatory and inhibitory synaptic proteins are reduced in AD and DS. a, c, e, g** Plots showing the proportion of PSD-95+ and gephyrin (GPHN) + puncta in each layer with immunofluorescence in low (<75), mid (75–90), or high (>90) intensity ranges for each group (group means ± SEM; based on Fig. 1e-h IR (Immunoreactivity) intensity distributions). Each dot represents the value for one subject. RM-ANOVA (group × intensity interaction) indicated a significant effect for layer 2 only: $P = 0.0019$ for PSD-95 and $P = 0.0133$ for GPHN. RM-ANOVA for layer 1, $P = 0.1027$ for PSD-95 and $P = 0.4151$ for GPHN ($n = 5$ controls, 5 AD and 6 DS). **b, d, f, h** Box plots show the ratio of high-to-low labeling intensities for each individual case, the median is represented by the line within the box, and the first and third quartiles are represented by the ends of the box. The whiskers extend from each end of the box to the first or third quartile ±1.5 (interquartile range). As shown, all AD and DS cases exhibited significantly lower high-to-low intensity ratios as compared to controls for both postsynaptic proteins in each layer: PSD-95 layer 1, ($F$ (2,13) = 14.17, $P < 0.0005$); GPHN layer 1 ($F$ (2,13) = 5.1, $P = 0.023$); PSD-95 layer 2, ($F$ (2,13) = 8.4, $P = 0.0045$); GPHN layer 2 ($F$ (2,13) = 37.54, $P < 0.0001$). $P$ values shown are from the two-sided Dunnett's post hoc test comparing AD or DS to control.

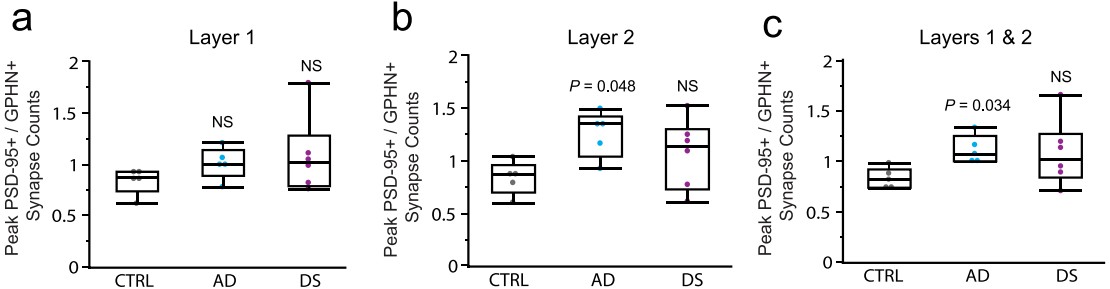

**Fig. 3 Differential alterations in the excitation to inhibition balance in parietal cortex of AD and DS.** Boxplots show for each subject the ratio of the peak value from the PSD-95 immunopositive (+) synapse intensity frequency distribution to the peak value from the GPHN + synapse distribution to provide a measure of the anatomical excitatory to inhibitory (E/I) balance for layers 1 (**a**) and 2 (**b**), and for the two layers combined (**c**) ($n = 5$ controls, 5 AD and 6 DS). As shown, the E/I balance (PSD-95/GPHN ratio) was not different between groups in layer 1 (**a**; Welch ANOVA allowing unequal variances $P = 0.22$; NS = not statistically significant), but it was significantly affected in layer 2 (**b**; Welch ANOVA; $P = 0.0316$) with only the AD group exhibiting elevated E/I balance versus controls ($P = 0.048$, nonparametric two-sided Dunn's post hoc test). Combining layers 1 and 2, the difference between the AD and control groups was even greater (Welch ANOVA $P = 0.0173$; AD vs CTRL, $P = 0.0336$ two-sided Dunn's posthoc test). Notably, the DS group was highly variable in these E/I balance analyses. The median is represented by the line within the box, and the first and third quartiles are represented by the ends of the box. The whiskers extend from each end of the box to the first or third quartile ±1.5 (interquartile range).

electrophysiological E/I ratio for each subject. For this, measures were collected and averaged only from oocytes in which both kainate and GABA currents were measured as described in methods. As compared to controls, only the AD group exhibited an increase in electrophysiological E/I balance with the kainate/GABA response ratio being significantly greater than for controls ($F (2,13) = 4.299$, $P < 0.0369$; Fig. 5f); responses from the DS group did not differ from controls. Combined with results from the FDT and flow cytometry analyses, these findings provide strong evidence that individuals with AD have a shift in the synaptic E/I balance leading to greater excitatory relative to inhibitory synaptic activity in cortex than is the case in controls.

Unlike the AD cases, the electrophysiological E/I ratio was less affected in DS (vs controls) despite these individuals exhibiting severe global plaque and tangle pathology (Table 1 for averages and Supplementary Data 1 for details), and similar plaque load in parietal cortex comparable to the AD group (Fig. 1b). As described above, the DS group exhibited large variability in ion currents that lead us to ask whether this might reflect another aspect of pathology such as levels of phosphorylated (p) Tau, which can vary across individuals despite similar AD staging[26]. Thus, we tested if ion currents in the DS group were correlated with pTau levels (Supplementary Data 2). Notably, in DS individuals the amplitude of kainate currents was negatively correlated with pTau levels ($\rho(6) = -0.9429$, $P = 0.0048$; non-parametric Spearman's $\rho$ to avoid artifactual correlations due to extreme variability). In addition, the numbers of synapses with high levels of PSD-95-ir (layers 1 and 2 combined) were also negatively correlated with AT8-ir denoting pTau levels ($\rho (6) = -0.8857$, $P = 0.0188$). AT8-ir had a lower correlation with the amplitude of GABA currents ($\rho = -0.886$, $P = 0.02$) than with kainate currents and was not correlated with GPHN-ir per synapse ($\rho = -0.09$, $P = 0.87$). For the AD subjects, pTau levels did not correlate with either measure ($P > 0.2$ for all; Supplementary Data 3) likely reflecting the small within-group difference in pTau levels.

**Increased transcriptomic DLG4/GPHN ratio in AD.** Using several independent approaches on the same set of subjects, the above results demonstrate that while both excitatory and inhibitory synapses are affected in AD, the E/I balance of postsynaptic elements is markedly elevated in this disorder relative to the controls. To further determine whether the E/I imbalance in AD is reflected at the transcriptional level in an independent cohort,

we assessed the publicly available RNA-Seq data from the ADTBI study[27] to estimate the transcriptional E/I ratio in PCx. The analysis included the eight cognitive healthy controls, between 78 and 100 years of age with no history of neurodegenerative or psychiatric disorders, and 12 subjects with a DSM-IV clinical diagnosis of dementia of the AD disease type with similar age and high levels of AD neuropathology, that have similar neuropathological characteristics to the cohort used in FDT and MSM analyses (Supplementary Data 4 for demographics)[28,29]. The transcriptional E/I ratio, defined as the level of mRNA for DLG4, which encodes PSD-95, divided by the level of mRNA for GPHN, which encodes gephyrin, was significantly elevated in AD ($P = 0.017$, nonparametric Wilcoxon/Mann–Whitney $U$-test) (Fig. 6a). The difference in the DLG4/GPHN ratio was driven by a reduction in GPHN ($P = 0.0087$), but not DLG4 ($P = 0.82$), in subjects with AD.

Because the relationship of transcript level and protein concentration is gene-specific and not necessarily linear[30] the implication that changes in the DLG4/GPHN ratio represent synaptic alterations is not granted. Therefore we used three additional analyses to determine whether alterations in this ratio represent synaptic alterations: (1) multivariate correlation analysis of the transcriptome in the parietal cortex with the DLG4/GPHN ratio followed by gene ontology (GO) of top correlated genes to determine cluster representation, (2) fold change of gene expression in predetermined excitatory and inhibitory GO modules and (3) weighted gene co-expression analysis (WCGNA) to determine unbiased modules of gene co-expression. The DLG4/GPHN ratio was positively correlated with 1317 genes at the level of $P < 0.01$ (Supplementary Data 5). GO analysis of these genes identified the postsynapse (GO:0098794) as the most representative category for cellular component where these genes belong (Supplementary Data 6), indicating that changes of the DLG4/GPHN ratio in AD reflect broad alterations at the postsynaptic level in this cohort (Fig. 6b). Our second analysis of differential gene expression relative to standardized a priori GO modules[31,32] confirmed the downregulation of GPHN, which was also the gene with the lowest p-value among the preselected excitatory/inhibitory genes. Interestingly, it also showed changes of the expression of excitatory and inhibitory genes in both directions suggesting a complex pattern of remodeling of synaptic proteins combined with changes at the cellular level (Supplementary Fig. S3; Supplementary Data 7 and 8

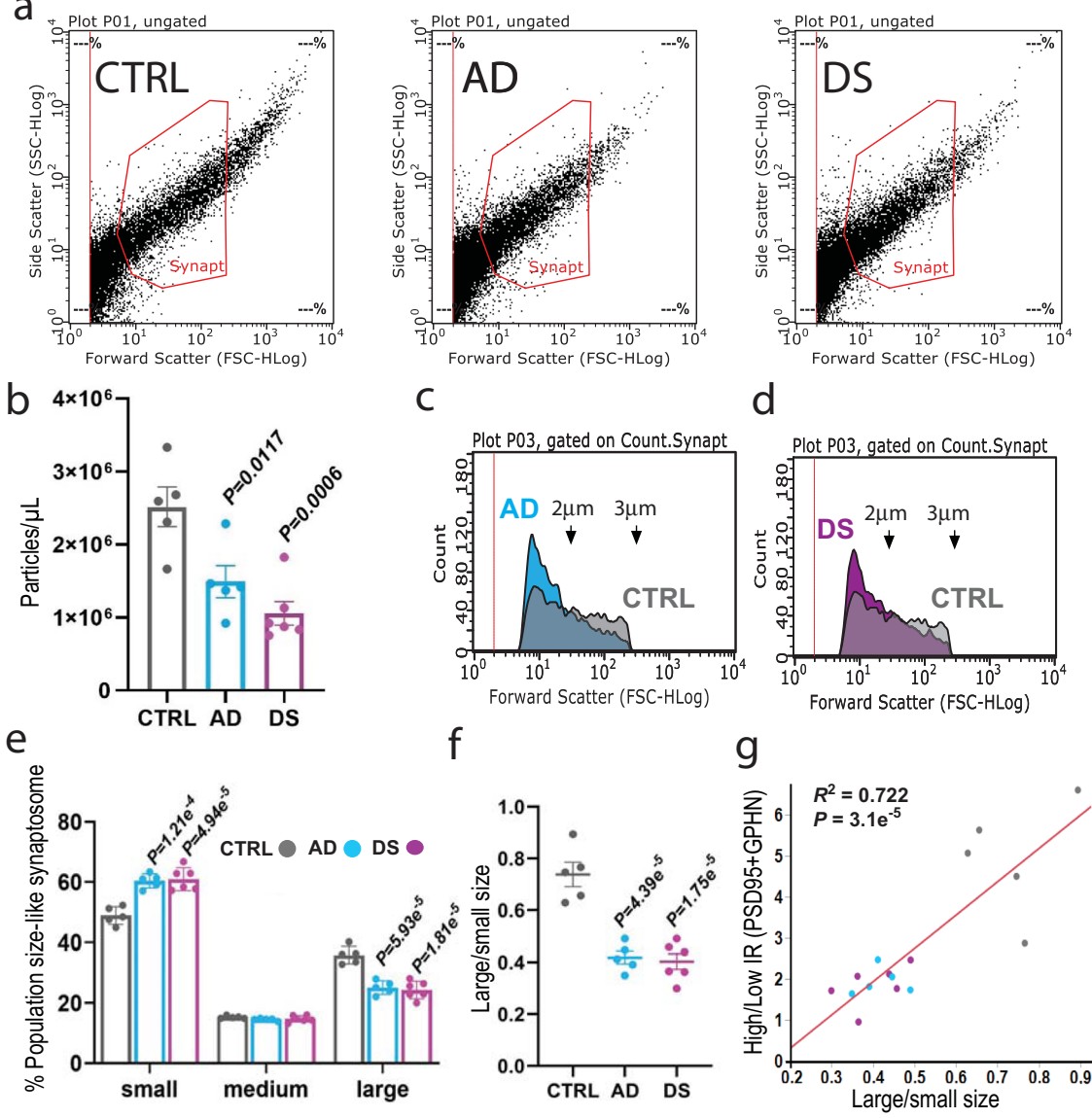

**Fig. 4 Flow cytometric analysis of synaptosomes-like particles in P2 fractions from AD and DS subjects. a** Representative plots indicate the gating parameters, based on size (1–3 µm), used to quantify synaptosomes in P2 fractions from the parietal cortex of CTRL, AD, and DS subjects. **b** Numbers of particles within the size of synaptosomes were reduced from $2.5e^{+6} \pm 2.7\,e^{+5}$ in control (mean ± SEM; $n = 5$ biologically independent samples, 1 from each control subject, each value per subject is the average of two independent experiments) to $1.5\,e^{+6} \pm 2.2\,e^{+5}$ in AD ($n = 5$ samples, 1 from each AD case) and $1.6\,e^{+6} \pm 1.6\,e^{+5}$ in DS ($n = 6$ samples, 1 from each DS case). One-way ANOVA showed effect of diagnosis on particle count was significant, $F(2,13) = 12.04$, $P = 0.0011$. Post hoc analysis using two-sided Dunnett's method (AD and DS vs CTRL) indicated that the average number of particles was lower in AD and DS as compared to control ($P$ values in the figure). **c, d** Representative forward scatter plots for synaptosome-sized particles from an AD and a DS subject compared to the same control. Notice a shift to smaller size particles in AD and DS subjects compared to the control. **e** Plots showing the proportion of particles within the size range of synaptosomes in P2 fractions that were further identified as being small (1 µm < diameter (∅) < 2 µm), medium (2 µm = ∅) or large (2 µm < ∅ < 3 µm) sized for each group (group means ± SEM; based on **c, d** size distributions. See Supplementary Fig. S2 for gating strategy). One-way ANOVA determined that effect of diagnosis was significant for large ($F(2,13) = 29.00$, $P = 1.61\,e^{-5}$) and small-sized particles ($F(2,13) = 24.30$, $P = 4.06\,e^{-5}$) but not for the medium group ($F(2,13) = 1.575$, $P = 0.244$). **f** All AD and DS cases exhibited significant reductions (one-way ANOVA) in the large-to-small size ratios for particles from P2 fractions ($F(2,13) = 29.83$, $P = 1.4\,e^{-5}$). $P$ values on bars are from two-sided Dunnett's post hoc test comparing AD or DS vs control. **g** Plot showing the correlation between high-to-low immunoreactivity (IR) ratios for all FDT analyzed synapses (data for both PSD-95-ir and GPHN-ir in layers 1 and 2 were combined) and the large-to-small synaptosomal particles for each subject. Each subject is color coded by diagnosis: CTRL (gray), AD (cyan), DS (magenta). The solid line represents the linear fit; $R^2(16) = 0.72$, $P = 3.1\,e^{-5}$, showing agreement between intensity and size data from FDT and flow cytometry driven by group separation.

for GO modules and differential expression details). Our third analysis using WCGNA found 30 modules of co-expression in this cohort (Supplementary Data 9), of which 3 were differentially expressed in AD vs controls ($P < 0.05$; Wilcoxon; Supplementary Fig. S4). One of these modules (ME20) is enriched for "neuron projection membrane" (GO: 0032589), and "axon part" (GO: 0033267), and the other two (ME25 and ME26) converge onto the perinuclear region of the cytoplasm (GO: 0048471). In

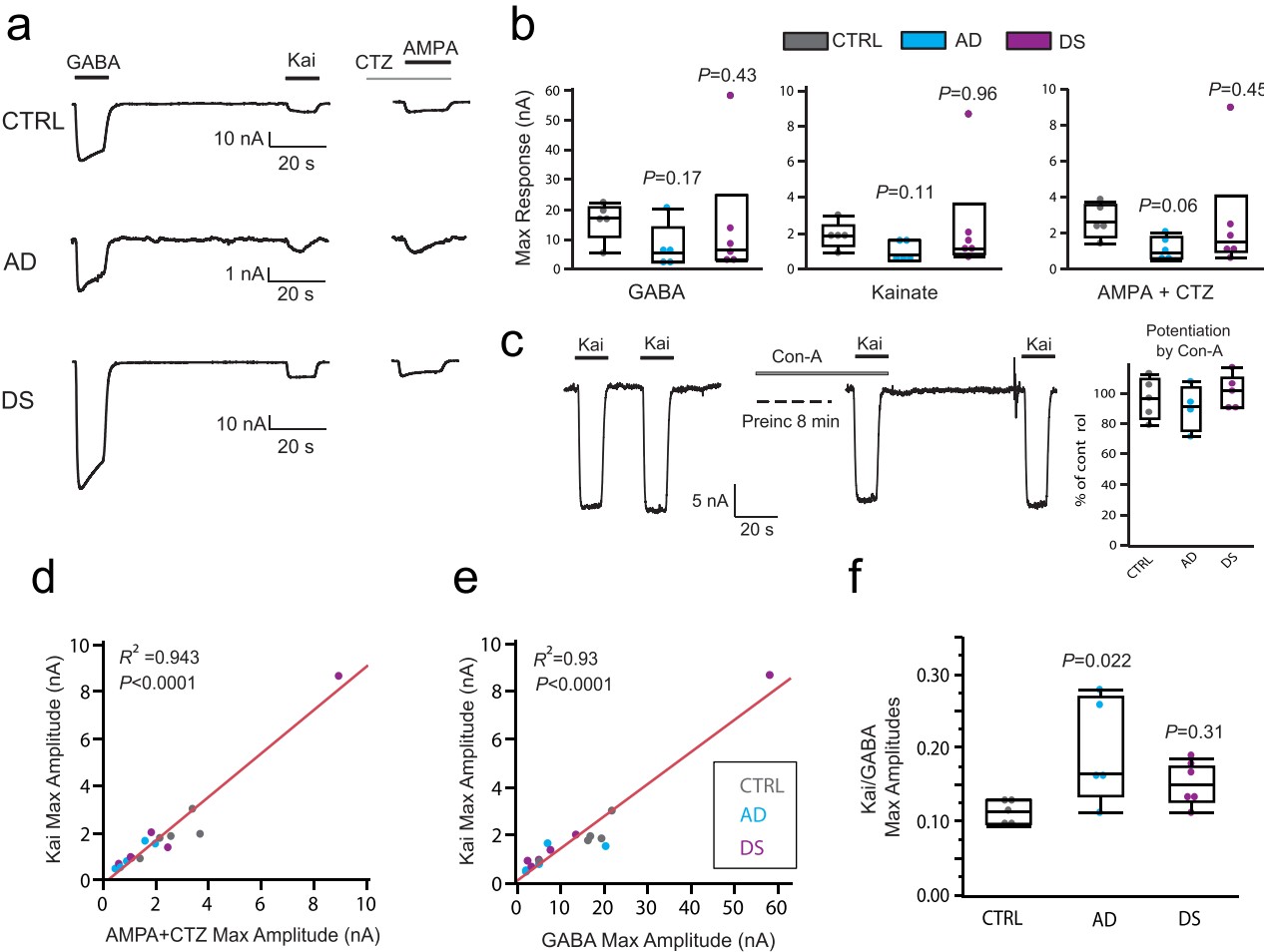

**Fig. 5 Differential alterations of the excitation to inhibition balance in ion currents of microtransplanted receptors from AD and DS parietal cortex.**
**a** Functional responses of microtransplanted synaptic receptors were elicited by application of 1 mM GABA, 100 μM kainate (Kai), or 10 μM s-AMPA in combination with 10 μM cyclothiazide (CTZ), after 3 min preincubation with CTZ. **b** No significant group differences were found in the maximal amplitude of agonist-induced ion currents for any of the responses (non-parametric Wilcoxon test $P = 0.13$ for kainate, $P = 0.08$ for AMPA + CTZ, $P = 0.18$ for GABA; $n = 5$ controls, 5 AD and 6 DS). $P$ values in each plot are from two-sided Dunn's post hoc rank test comparing groups versus control. The DS group had both the largest, and some of the smallest, responses indicating a large within-group data spread in this condition. **c** Kainate-induced responses were not potentiated by concanavalin A (Con-A), a positive allosteric modulator specific for kainate-type glutamate receptors, and did not show differences between groups (one way ANOVA ($F$ (2,11) = 0.66, $P = 0.5$; $n = 5$ controls, 5 DS and 4 AD), indicating that the participation of kainate receptors is negligible and kainate-induced receptors are generated by AMPA-type receptors. **d,e** Plots showing the correlation between the maximum amplitude responses of excitatory receptors activated by kainate or AMPA plus CTZ and those of inhibitory receptors activated by GABA. Each point represents the average of responses from multiple independent recordings in oocytes from 7 frogs. For the exact number of oocytes per subject in all MSM experiments see Supplementary Data 2. The solid lines represent the Pearson's correlation. **f** Excitation to inhibition ratio, defined as the average of maximum amplitude of kainate- to GABA-induced currents measured from the same oocyte was significantly larger in oocytes microtransplanted with synaptic membranes from AD compared to control brains ($F$ (2,13) = 4.3, $P = 0.037$; $n = 5$ controls, 5 AD and 6 DS). $P$ values in plot are from two-sided Dunnett's post hoc test comparing groups versus control. For all box plots, the median in each group is represented by the line within the box, and the first and third quartiles are represented by the ends of the box. The whiskers extend from each end of the box to the first or third quartile ±1.5 (interquartile range). Outliers are represented beyond the whiskers.

summary, the three analyses converge to show a bias for reduced gephyrin expression in AD as well as more general changes at the cellular level.

**Elevated excitatory to inhibitory neuronal ratio in AD.** To further elucidate sources of E/I disruption in the parietal cortex in AD, we used publicly available in situ hybridization image data from the ADTBI study[27], to quantify the number of glutamatergic neurons that expressed mRNA for the excitatory vesicular glutamate transporter 1 (vGluT1) and GABAergic neurons that expressed mRNA for GABA transporter 1 (GAT1), in a well-defined area of the cortex. For this, vGluT1 mRNA+ and

GAT1 mRNA+ cells in parietal cortex layers I-VI were counted (Fig. 7a, b) and their densities (number of cells/ area) in AD were compared to controls. Densities of vGluT1 mRNA+ cells were not different between groups ($P = 0.6$, Wilcoxon test) (Fig. 7c). In contrast, densities of GAT1 mRNA+ cells were significantly reduced in AD ($P = 0.015$; Fig. 7d). Comparing the ratios of vGluT1 to GAT1 expressing cells for each case demonstrated a marked increase in this cellular E/I ratio in the AD group ($P = 0.0026$; Fig. 7e and Supplementary Fig. S5). Importantly, the DLG4/GPHN ratio that represents postsynaptic elements (Fig. 6b) was linearly correlated with the E/I cell ratio, and the correlation was driven by changes in the AD group

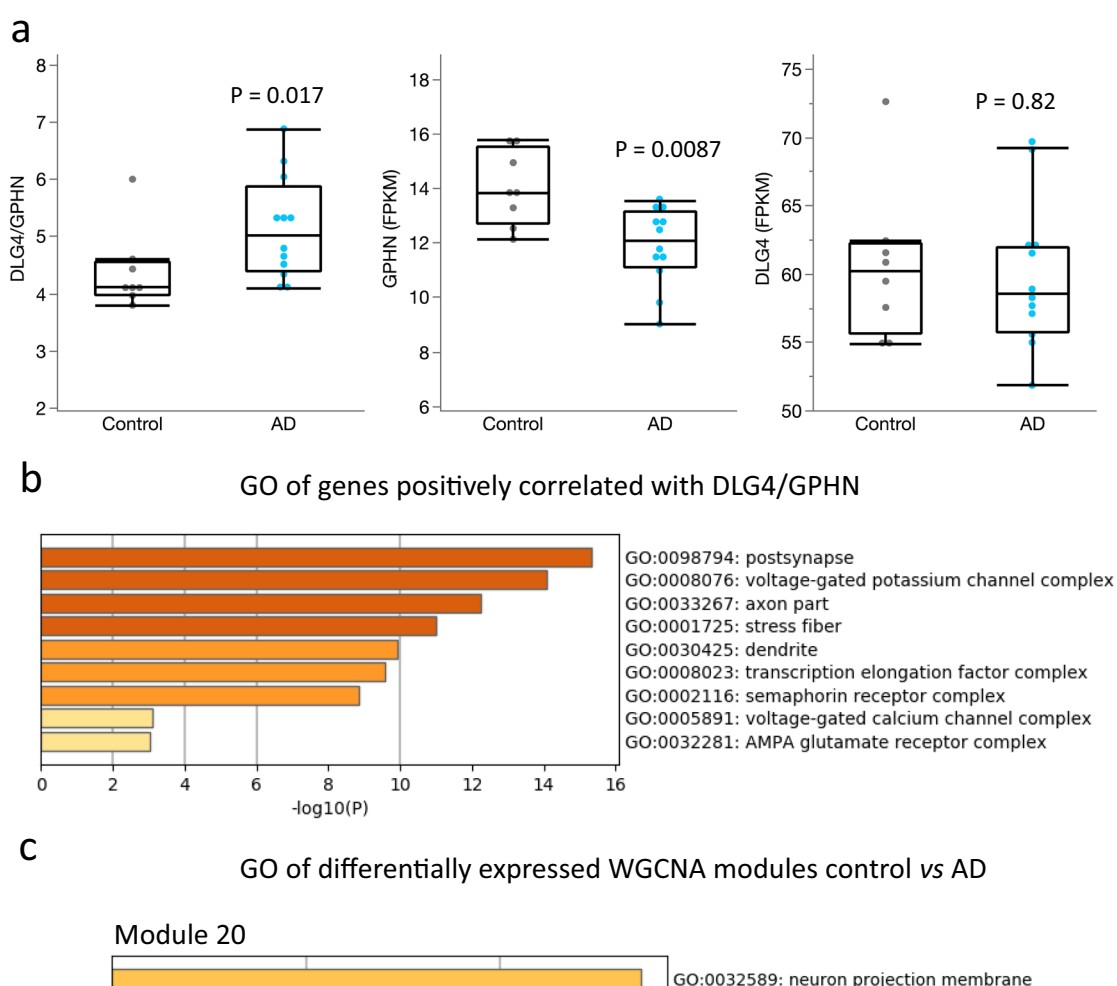

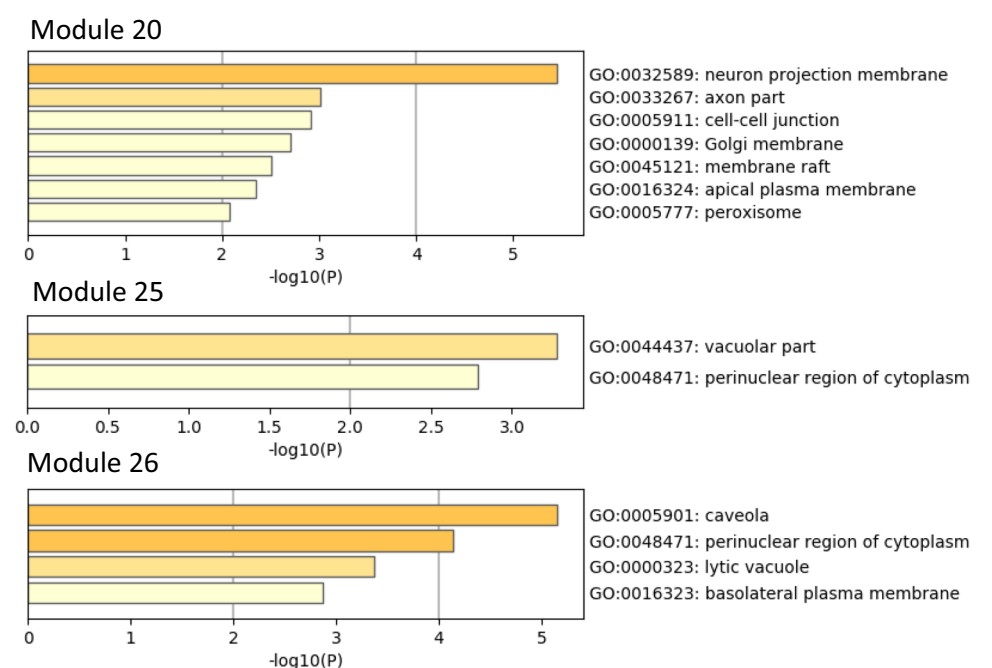

**Fig. 6 Differential alterations in expression of excitatory and inhibitory postsynaptic density proteins are associated with dementia in AD. a** The transcriptional E/I ratio for DLG4 to GPHN (DLG4/GPHN) using RNA-Seq datasets from the Aging, Dementia and Traumatic Brain Injury study (ADTBI), was significantly increased in AD cases compared to controls with no pathology ($n = 8$ cognitive healthy controls, CERAD = 0 and 12 subjects with a DSM-IV clinical diagnosis of dementia of the AD disease type and AD pathology CERAD = 3). The difference was driven by reduction in GPHN expression and not from DLG4, *P* value from the non-parametric two-sided Wilcoxon test. **b** Top representative clusters of the gene ontology (GO) analysis for cellular component of genes positively correlated with the DLG4/GPHN ratio implemented in Metascape. **c** Top representative clusters of the gene ontology analysis for co-expression modules differentially expressed between AD and control cases. See Supplementary Data 5–9 for details in gene correlations, GO enrichment, differential expression, and module membership.

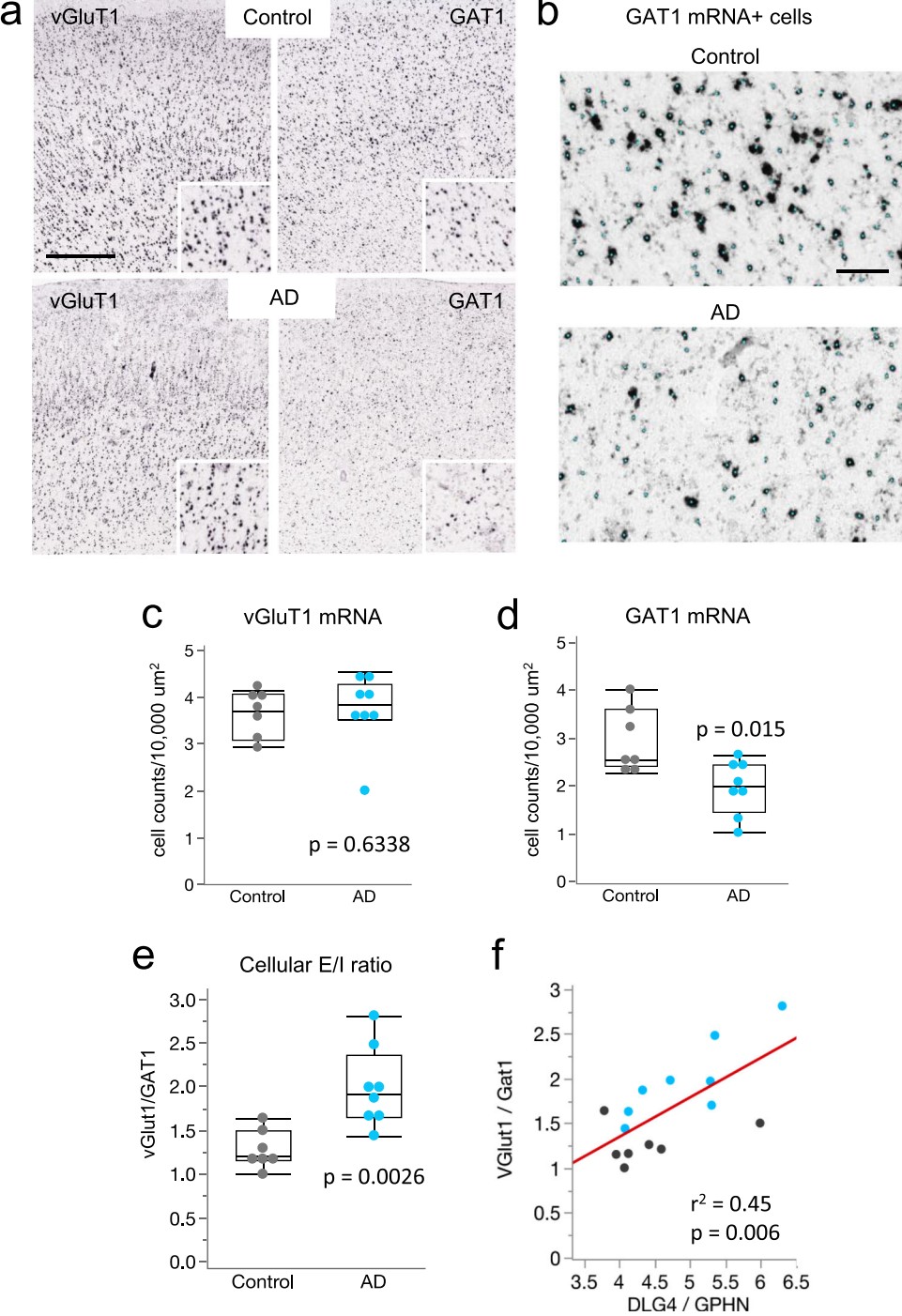

**Fig. 7 Marked reductions in GAT1 mRNA expressing cells in AD cases with dementia. a** Images showing colorimetric in situ hybridization for vGluT1 and GAT1 mRNAs in a control no dementia-CERAD 0 case and a dementia-CERAD 3 AD case. Insets show labeled cells at higher magnification. Calibration bar, 750 μm; 375 μm for inset. **b** Images showing examples of GAT1 mRNA positive (+) cells that were identified and counted in the analyses; identified cells are overlaid with a small cyan dot. Calibration bar, 100 μm **c,d** Quantification of vGluT1 and GAT1 mRNA expressing cells in parietal cortex per 10,000 μm$^2$ (plots show median values, 25th, and 75th percentiles, and minimum and maximum range). Data shown are based on a 10 μm$^2$ size threshold: separate analyses using 30 μm$^2$ size threshold yielded similar differences in E/I ratios between groups (Supplementary Fig. S5). GAT1 expressing cells were reduced in the AD dementia-CERAD 3 group ($n = 8$ AD subjects) versus the control, no dementia-CERAD 0 group ($n = 7$ healthy controls). *P* values are from the two-sided Wilcoxon Test. **e** vGluT1/GAT1 ratios for the same subjects; individual vGluT1 to GAT1 comparisons are presented in Supplementary Fig. S5. The cellular E/I ratio was significantly elevated in the AD group versus controls (**$P < 0.01$ two-sided Wilcoxon test; $n = 7$ controls, 8 AD). **f** Linear correlation between the cellular (vGluT1/GAT1) and transcriptional (DGL4/GPHN) expression E/I ratios for cases with both data sets (see Supplementary Data 6); AD ($n = 8$) cases, cyan; control cases ($n = 7$), black.

(Fig. 7f and Supplementary Fig. S6). These results support the conclusion that the synaptic changes in PSD95 and GPNH in AD are among more widespread cellular changes including marked decreases in GABAergic neurons expressing GAT1 that contribute to the shift in E/I balance in the parietal cortex of AD patients.

## Discussion

Using multiple approaches, we report here evidence for a shift of the global E/I synaptic balance in the inferior PCx of the AD brain that favors greater synaptic excitation. While results from FDT studies indicate a reduction in levels of both excitatory and inhibitory synapse markers of early-onset AD subjects, the ratio of PSD-95 to gephyrin was largely altered in layer 2, indicating greater excitatory connectivity in this region. Notably, global synaptic E/I ratio positively correlates with neuronal firing[33], and while we cannot exclude the possibility that these synaptic E/I effects might reflect a feature of early-onset AD such as seizure activity[34,35], complementary findings from the ADTBI cohort suggest that E/I imbalance is not an exclusive feature of early-onset AD. In particular, gene expression analyses for both PSD-95 and GPHN transcripts and vGluT1 and GAT1 expressing cells in the ADTBI cohort of subjects, with late-onset AD and level of pathology similar to the ones used in FTD studies, also demonstrated a shift in E/I imbalance. Importantly, these findings were based on analyses of both superficial and deep layers, suggesting that the synaptic protein alterations in layer 2 may extend to other cell-dense layers in the PCx. Electrophysiological recordings from the MSM studies also showed unequal deficits in postsynaptic AMPAR- and GABA_AR-mediated ion currents, leading to an increase in the global electrophysiological synaptic E/I ratio, which would favor enhanced synaptic excitatory drive. The E/I shift in AD is remarkable in the context of studies showing that synaptic levels of inhibition are generally proportional and scaled in strength to excitation, despite even large variations in the amplitude of excitatory synaptic currents across neurons[10,36,37]. Changes in excitatory synapse number and/or strength predicted for Hebbian plasticity during learning and memory are similarly compensated for by synaptic scaling[38], heterosynaptic plasticity[37], and changes in synaptic function within minutes and hours[39]. Indeed, the E/I ratio in our MSM study had minimal variation compared to ion currents measured individually, indicating that on average the relationship between postsynaptic AMPARs and GABA_ARs is nearly constant in control individuals, and even in DS.

While our results do not identify the cell types receiving higher excitatory drive in AD, the large majority of neurons in cortex are excitatory pyramidal cells[40], suggesting that these are the principal target. Consistent with this, there is enhanced excitatory transmission in principal neurons due to GABAergic deficits in animal models of familial AD[4,41,42]. Interneurons are somewhat protected compared to pyramidal neurons in AD at early stages, but nonetheless exhibit morphological alterations and innervation deficits[43], particularly adjacent to amyloid pathology[44], that are likely to underlie the beginning of synaptic inhibitory dysfunction in the disorder. Based on our gene expression findings for advanced AD, we propose that as the disease progresses there are more cell-wide effects on inhibitory neurons, which would be expected to further exaggerate E/I deficits.

The cellular mechanisms contributing to the emergence and progression of AD are poorly understood. Multiple studies have described dysfunctions in cholinergic and glutamatergic systems[45,46] but it is not known if equivalent disturbances occur in the GABA system. Post-mortem AD studies of GABA levels, or the distribution or activity of the GABA synthesizing enzyme glutamic acid decarboxylase (GAD), have obtained mixed results,

with reports indicating reductions or no change in GABA or GAD levels (see[47] for a comprehensive review). Analyses of GABA_ARs in the AD brain have reported decreases in protein levels for the α1, α5, β3, and γ2 subunits[47], and preservation or up-regulation of subunits that are normally co-expressed in the human brain[48,47]. Total levels of GPHN immunoreactivity are reported to decrease with increasing pathology in the AD brain[49] and there is evidence for a transcriptional decoupling between GABA_AR subunits and GPHN[12], two effects that would be consistent with inhibitory deficits in AD. Of interest in this regard is evidence that gephyrin becomes abnormally associated with Aβ plaques and occasionally with NFTs[50]. However, no studies to date have evaluated synaptic levels of GPHN in AD. Thus, the present finding demonstrating that the density of gephyrin-ir in synapse-size puncta is more greatly reduced than those for PSD-95-ir in AD parietal cortex indicates a relatively greater deficit at inhibitory synapses and thus an increase in "uncompensated" excitatory connectivity. The electrophysiological E/I ratio in our samples, and the transcriptional E/I ratio in an independent cohort, also indicates larger deficits in inhibitory postsynaptic receptors. Importantly, whether the total gene expression for gephyrin is affected by abnormal extrasynaptic expression in AD (association with plaques[50] and dystrophic processes as seen in the present study) is not known. A limitation of our transcriptional analysis is the lack of corresponding measures for PSD-95 and gephyrin protein content in the same cohort, as these data are not available. However, we did measure the expression of PSD-95 and gephyrin by immunohistochemistry and FDT in our first cohort, thus cross-validating the increase in the PSD-95/GPHN transcription ratio at the protein level specifically in synapses by FDT. Future studies combining RNAseq with anatomical and functional synaptic measures in the same cohort will provide additional information on the predictive value of transcriptomic and synaptic dysfunction.

Our study was designed to characterize the relationships between excitatory and inhibitory synaptic changes, from the anatomical to the electrophysiological domains, in a small cohort. Therefore, it was not sufficiently powered to detect changes in electrophysiological responses of synaptic receptors measured individually, wherein the variance across cases can be substantial. Despite this difference in absolute current levels, AMPAR and GABA_AR mediated responses were highly correlated with an E/I ratio that was significantly elevated in the AD group. The present work also tested whether E/I measures differed between individuals with AD alone and those with DS plus AD pathology. While the DS cases exhibited marked reductions in PSD-95 and GPHN immunofluorescence, similar to the AD group, the anatomical and electrophysiological E/I ratios for the DS group were not significantly different from controls, indicating that either homeostatic mechanisms are still functional or disturbances to excitatory and inhibitory synapses are comparable. Unlike the AD cases, the DS group exhibited greater variability in immunocytochemical and electrophysiological measures for excitatory synapses that were related to pTau levels. Specifically, numbers of high-density PSD-95+ puncta and the amplitudes of kainate-induced AMPAR currents were both negatively correlated with AT8-positive (Ser202 and Thr205) pTau levels. These findings are consistent with previous work showing that tau and PSD-95 interactions at synapses are reduced following phosphorylation at the AT8 site with tau being shifted towards interactions with the Src-family tyrosine kinase Fyn[51]. An increased association with Fyn may play a role in tau-mediated neurotoxicity[52], whereas prolonged phosphorylation of tau can lead to reduced AMPAR-dependent synaptic transmission[53].

Our converging evidence suggests that a pro-excitatory synaptic imbalance may underlie the hyperexcitability and

reduced resting state deactivation that have been consistently observed in the PCx of AD patients[54,55]. Thus, cortical hyper-excitability may be part of a positive feedback loop whereby higher neuronal activity promotes production and accumulation of toxic Aβ and Tau oligomers[56] resulting in unequal synaptic losses that, in turn, lead to greater excitation. The preservation of a normal synaptic E/I balance in PCx in DS is an interesting finding. To our knowledge, it is not known whether the default mode network shows signs of hyperactivity in DS. However, comparative imaging studies of DS and control subjects did not find functional differences in the resting-state connectivity within PCx, even in subjects with *APOE* ε4 vs *APOE* ε3 variants that denote a higher risk for AD[57], suggesting resilience or persisting homeostatic mechanisms in this region in DS despite the burden of ADNC. Further studies aimed at understanding the molecular and electrophysiological basis of other hyper- and hypo-functional regions in late onset AD brain and related dementias should be informative about the remodeling changes at the synaptic level and may open more specific ways to correct the synaptic imbalance.

In conclusion, measures of excitatory and inhibitory synapses are highly correlated at the anatomical and electrophysiological levels in PCx of control subjects in middle age, and even in DS, a disorder in which synapse loss is pronounced. In contrast, increases in the synaptic E/I ratio in AD indicate a failure of homeostatic mechanisms leading to increases in relative levels of excitatory synaptic drive that contribute to cortical hyperexcit-ability and cognitive impairment in AD. Further studies aimed at elucidating whether E/I ratios in AD are associated with different types to dementia and identifying mechanisms that can restore E/I balance, could prove invaluable for devising useful treatments to offset cognitive decline in this devastating disorder.

## Methods

**Cases and tissue samples**. AD and DS parietal cortex tissue was provided by the UCI Institute for Memory Impairments and Neurological Disorders through the UCI-Alzheimer's Disease Research Center (ADRC). Control tissue was provided by the NINDS/NIMH sponsored Human Brain and Spinal Fluid Resource Center (HBSFRC) at the VA West Los Angeles Healthcare Center, Los Angeles, CA (Supplementary Data 1). Informed consent was obtained from all the participants and the use of the material, clinical and demographic information had been obtained by the ADRC and the HBSFRC. All cases were de-identified and coded by the source tissue banks following Health Insurance Portability and Accountability Act (HIPAA) policy for the use and publishing of demographic information shown in Supplementary Table S1. For the present studies, the samples were recoded and processed for all analyses with the experimenter blind to subject and group. The study was reviewed by the Institutional Review Boards of the University of California at Irvine and the University of Texas Medical Branch and categorized as a non-human subject study.

Postmortem human PCx tissue blocks, that included a well-defined sulcus with portions of gyri present on both sides, were obtained from five controls, five AD cases, and six DS cases, with both males and females included in the analyses (Table 1). While effort was made to match age and postmortem interval (PMI) as closely as possible, the control group contained longer PMIs; in other studies by our group, PMIs up to ~25 h did not significantly affect FDT- or MSM- based synapse measures[20,58,59]. In addition, RIN values for the controls ranged between 6.6 and 8.6 (median = 7.9) indicating relatively good tissue quality. Gross and microscopic evaluations of the control brains were conducted by a neuropathologist at the HBSFRC and, in all cases, there was no diagnostic evidence of neuropathology, hypoxia or metastatic disease; all controls were diagnosed as essentially normal brains. All DS and AD cases underwent formal neuropathological evaluation in the ADRC Neuropathology Core and demonstrated Braak stage VI neurofibrillary degeneration (distribution of neurofibrillary tangles)[60] and Consortium to Establish a Registry for Alzheimer's Disease (CERAD) frequent cerebral cortex neuritic plaque density[61]. Functional levels of independence were comparable between the two groups as determined by the modified version of the Bristol Activities of Daily Living Scale that was collected within the last 1–2 years of life[62] (Table 1). In addition, in a recent study, we used western blotting and antibodies to AT8 to assess the levels of hyperphosphorylated (p)-tau in tissue sections obtained from the same tissue blocks used here[58]. Those data are presented here again for each sample in order to directly compare a hallmark of AD pathology with measures of excitation and inhibition for each case.

For both FDT and MSM analyses, each fresh-frozen tissue block was cut on a cryostat at 20 μm perpendicular to the tissue surface and sections were collected as illustrated in Fig. 1a. For FDT, sections were slide-mounted, methanol-fixed, and stored at −20 °C prior to processing for double-labeling immunofluorescence. For MSM analyses, a single tissue section was collected into a microcentrifuge tube and stored at −80 °C until processed as described below. Additional slide-mounted sections from all cases were also stained and assessed for the presence of amyloid plaques as described below.

**Amyloid plaque staining**. Slide-mounted tissue sections from all cases (AD, DS, controls) were fixed in 4% paraformaldehyde (in 0.1 M phosphate buffer, pH 7.2) and processed together using the Amylo-Glo RTD Amyloid Plaque Stain Reagent (Biosensis, Australia) according to manufacturer instructions. Amyloid plaque staining was visualized under UV illumination. For quantification, images (each 574,200 μm$^2$) were collected using a 10× objective on a Leica DM6000 epi-fluorescence microscope equipped with a Hamamatsu ORCA-ER digital camera; 20 images per case were taken along the sulcus at spaced intervals. Coded images were assessed for plaques and counted by hand, blind to the condition. Numbers of plaques across the total area examined per case were normalized to counts per 0.5 mm$^2$. Differences between AD and Ds groups were assessed by unpaired Student's t-test; controls lacked staining (verified by visual assessment of entire sections) and therefore were not included in statistical analyses.

**Fluorescence deconvolution tomography (FDT)**. Tissue sections were processed for double-labeling immunofluorescence and FDT as described[63–65], using antisera directed against the excitatory synapse scaffold protein PSD-95 to label excitatory synapse PSDs[66] in combination with antisera directed against the inhibitory synaptic marker gephyrin. Briefly, slide-mounted tissue was washed (three times, 10 min each) in 0.1 M phosphate buffer (PB), incubated in primary antisera overnight at 4 ºC, washed in PB, and incubated in secondary antisera for 1 h at room temperature. After a final wash in PB, tissue was coverslipped using Vec-tashield with DAPI (Vector Laboratories, H-1200-10). Antibodies used were mouse anti-PSD-95 (1:1000; Thermo Scientific, #MA1-045) and rabbit anti-GPHN (1:1000; Abcam, ab32206), with species-specific Alexa Fluor 488 (Thermo Fisher Scientific, #A21202) and Alexa Fluor 594 (Thermo Fisher Scientific, #A21207) conjugated secondary antibodies (1:1000 each) for visualization. For FDT, image z-stacks of layer 1 and layer 2 parietal cortex were collected using a 1.4 NA 63× objective through a depth of 2 μm with 0.2 μm steps using a Leica DM6000 epi-fluorescence microscope equipped with a Hamamatsu ORCA-ER digital camera; the sample field size for each cortical layer measured 42,840 μm$^3$. Both layers 1 and 2 were assessed for analyses because depth placement of the sample field, relative to the cortical surface, could be reliably replicated, and to compare a relatively cell sparse versus cell dense layer. Numbers of immunolabeled synapses in the two layers ranged from 25,000 to 30,000 per stack. For each layer, 10–12 image stacks from three sections per brain were taken from within the sulcus of each sample, and then processed for iterative deconvolution using Volocity 4.1 (PerkinElmer). Deconvolved images were analyzed using in-house software to quantify all labeled puncta within the size constraints of synapses as previously described[64,65,67,68]. Background staining variations in the deconvolved images were normalized to 30% of maximum background intensity using a Gaussian filter, and object recognition and measurements of immunolabeled puncta were automated using in-house software built using Matlab R2007, Perl v5.30.0, and C99 that allows for detailed analysis of objects reconstructed in 3D. Pixel values (8-bit) for each image were multiply binarized using a fixed-interval intensity threshold series followed by erosion and dilation filtering to reliably detect edges of both faintly and densely labeled structures. Object area and eccentricity criteria were then applied to eliminate elements, including lipofuscin granules, that do not fit the size and shape range of synaptic structures from the quantification. Counts of immunolabeled objects were averaged across sections to produce mean values per subject. For immunolabeling density frequency distributions, counts were expressed as a per-centage of all labeled PSDs (PSD-95 or GPHN). Statistical comparisons to identify the effect of group used a one-way ANOVA followed by Newman–Keuls Multiple Comparison test for post hoc paired comparisons (GraphPad Prism, Version 8.0). If variances across groups were not equal, as determined by Bartlett's test, then the Welch ANOVA or nonparametric Kruskal–Wallis test was used followed by Dunn's Multiple Comparison test. In other comparisons, repeated-measures ANOVA was used followed by Student Newman Keuls posthoc test. In all cases, $P < 0.05$ was considered significant. Pearson product moment was used for all the linear correlations using JMP version 14 (SAS Institute, Cary, NC).

**Microtransplantation of synaptic membranes and flow cytometry**. Membrane preparations were isolated from a single 20 μm slice of frozen PCx from each brain donor using Syn-PER method (Thermo Fisher Scientific); on average, the tissue blocks ranged in size from ~25–35 mm (length) by 15–20 mm (width). Briefly, a tissue slice was placed in an Eppendorf tube and stored at −80 °C until processed. To make synaptic membranes, each slice was suspended in 500 μl of Syn-PER extraction reagent, transferred to 2 mL glass/Teflon Dounce and stroked slowly for 15 times. The homogenate was then transferred to 1.5 mL Eppendorf and cen-trifuged at 1200 $g$ (4 °C) for 10 min. The supernatant (S1) was transferred to a new

Eppendorf and centrifuged at 15,000 g for 20 min. The pellet enriched in synaptosomes (P2 fraction) was resuspended in 15 µL of Syn-Per solution, aliquoted, and stored at −80 °C until further use for electrophysiology experiments. The amount of protein was determined using the Qubit protein assay reagent kit (Thermo Fisher Scientific). Numbers of particles within the size of synaptosomes were counted as described previously;[69–71] briefly, 1 µl of P2 fraction was diluted in 4 mL phosphate buffer solution, then 500 mL of this solution were loaded into a Guava EasyCyte flow cytometer (Guava Soft v2.7; EMD Millipore). A size gate was delimited using appropriate size standards (Spherotech, Inc) and 5000 events within the gate were analyzed using Guava Soft 3.3. For MSM experiments synaptosomes were sonicated in iced-water three times for 5 s, at 1 min intervals between sonications, to create small proteoliposomes that can fuse to oocytes' extracellular membrane. One day before electrophysiological recordings the synaptic membranes were injected into stage V–VI Xenopus laevis oocytes[12,59,72]. Each oocyte was injected with 50 nL of synaptic proteoliposomes (2 mg/mL protein concentration) and characterized 18–36 h post-injection. For harvesting of oocytes, Xenopus laevis (Xenopus I; Dexter, MI) were anesthetized in tricaine methane sulfonate (MS-222, 0.17%) and euthanized by decapitation in adherence to protocols approved by the institutional Animal Care and Use Committee of the University of California Irvine (IACUC-1998-1388) and the University of Texas Medical Branch (IACUC-1803024). The ovarian lobes were removed, cut into small pieces, and placed in Barth's solution [88 mM NaCl, 0.33 mM Ca(NO₃)₂, 0.41 mM CaCl₂, 1 mM KCl, 0.82 mM MgSO₄, 2.4 mM NaHCO₃, 10 mM HEPES (pH 7.4)] with 2 mg/ml collagenase type I (Sigma-Aldrich) for 2 h in constant rotation. After the enzymatic treatment, isolated stage V–VI oocytes were selected and maintained at 17 °C for the remainder of the experiment. Healthy-looking oocytes were injected approximately 24 h after enzymatic dissociation.

**Electrophysiological recordings**. Ion currents elicited by agonist perfusion were recorded by the two-electrode voltage clamp method between 18 and 36 h post injection[12]. Microelectrodes were filled with 3 M KCl and resistance of the microelectrodes ranged from 0.5–3.0 MΩ. Piercing and recording took place in a chamber (volume ≈ 0.1 ml) continuously perfused (6 mL/min) with Ringer's solution [115 mM NaCl, 2 mM KCl, 1.8 mM CaCl₂, 5 mM Hepes (pH 7.4)] at room temperature (19–21 °C). Oocytes were voltage-clamped to −80 mV. Ion currents were recorded and stored with WinEDR Ver 3.2.7 Strathclyde Electrophysiology Software (John Dempster, Glasgow, United Kingdom) and analyzed with pClamp Ver 11 (Molecular Devices, San Jose CA, USA). For the estimation of the E/I ratio we only included oocytes where GABA and kainate currents had clear activation and deactivation phases, a signal-to-noise ratio of at least 3:1 and were consistently activated by multiple applications in the same oocyte. Kainic acid, s-AMPA were purchased from Tocris (Minneapolis, MN). All other reagents were from Sigma (St. Louis. MO). Working solutions were made by diluting stock solutions in Ringer's solution. A total of seven frogs were used for MSM experiments. For all measures, for each subject, electrophysiological recordings were done at least in triplicate (three oocytes) in batches of oocytes from two to four different frogs, balancing the groups for equal number of subjects in each experimental run. Statistical comparisons to identify the effect of diagnosis used the mean of each metric, for each subject, as experimental unit in a one-way ANOVA, followed by post-hoc Dunnett's multiple comparisons versus control test (JMP, version 14). If variances across groups were not equal, then the Welch ANOVA or nonparametric Kruskal-Wallis test was used followed by Dunn's comparison vs control test. As a matter of confirmation, we also implemented a nested ANOVA with random effects mixed model, wherein the subjects were nested within diagnosis, and subject was tested as a random effect using the expected mean squares method. In all cases, P < 0.05 was considered significant. Pearson product-moment was used for linear correlations, and Spearman's rank for nonparametric correlations, using JMP version 14.

**Gene expression analyses**. The normalized RNA-Seq dataset for the inferior parietal cortex and demographic information for cases from the Aging, Dementia, and Traumatic Brain Injury (ADTBI) study[27] was downloaded from the website link (http://aging.brain-map.org/download/index). Persons who elected to participate in the brain autopsy program completed a consent packet that included: Consent for Autopsy, UW Neuropath Core Brain Aging and Neurodegeneration Brain Bank Autopsy and Tissue Donation to Brain Bank consent form, HIPAA authorization, and Autopsy Contact Information Sheet, which included consent to publish de-identified demographics and clinical information as shown in http://aging.brain-map.org/donors/summary and Supplementary Table S4. The analysis in our study consisted of eight cognitive healthy controls (five males and three females) between 78 and 100 years of age with no history of neurodegenerative or psychiatric disorders and no pathology (CERAD = 0) and 12 subjects with a DSM-IV clinical diagnosis of dementia of the AD type (six males and six females) in the same range of age and with frequent abundance of neuritic plaques (CERAD = 3). Subjects with multiple etiologies, or non-AD dementia of unknown or vascular origin, were excluded from the analysis. Dementia was diagnosed at consensus conferences using DSM-IV criteria and Alzheimer's disease as defined by NINCDS-ADRDA criteria[73]. The transcriptional E/I balance for each subject was calculated based on their ratio of the fragments per kilobase of transcript per mapped (FPKM) reads for PSD-95 transcript DLG4 to GPHN mRNA, and then

the effects of age, sex, and PMI on this measure were evaluated. No manipulation or transformation of the data was implemented for the calculation of this ratio (demographic information can be downloaded from the webpage or from Supplementary Table 4). We limited the derivation of the transcriptional E/I ratio to PSD-95 and GPHN to match our anatomical analysis in the first cohort. The overlap between the protein detected anatomically by FDT and the transcripts for the same proteins in a distinct cohort provides (1) cross-validation of data between the two cohorts, and (2) maintains the interpretation of our results within the context of the two most abundant and consistently expressed excitatory and inhibitory postsynaptic densities described in the literature. For statistical differences in gene expression in control versus AD subjects the nonparametric Wilcoxon/Mann–Whitney U-test test was used.

To determine enriched modules associated with the DLG4/GPHN ratio a response screening analysis using linear regression and controlling for multiple comparisons was done to find genes that linearly correlate with the DLG4/GPHN ratio. Positively correlated genes with p < 0.001 were selected for GO analysis of cellular components using whole transcriptome in the dataset as background, a p value cutoff of 0.01, minimum enrichment of 1.5, and gene prioritization by evidence counting, using Metascape[74].

For analyses of predetermined GO modules and WCGNA, only genes with FPKM > 2 in at least three samples (16,935 genes) were used for downstream analysis. WGCNA version 1.69[75], using R 4.0, was applied on log2(FPKM + 0.1) values using the blockwiseModules function and the following parameters: soft thresholding power of 8, the signed version of the topological overlap matrix, reassignThreshold of 0, mergeCutHeight of 0.25 and pamRespectsDendro set to FALSE. For the analysis of excitatory and inhibitory genes, 29 excitatory and 30 inhibitory neurotransmission-related gene ontology terms were manually chosen. Gene ontology annotations were downloaded from BioMart (ENSEMBL release 100). Genes present in both lists or low expression in PCx were then removed resulting in a total of 347 excitatory-related and 65 inhibitory-related genes. The average Log2 fold changes for each gene were then calculated by taking the linear regression slopes between the log2(FPKM + 0.1) value of each gene and clinical diagnosis (AD vs. control).

**In situ hybridization analyses**. Full-size high-resolution images of colorimetric in situ hybridization (ISH) for vGluT1 (SLC17A7) and GAT1 (SLC6A1) mRNA expressing cells in PCx were downloaded from the Aging, Dementia, and Traumatic Brain Injury (ADTBI) study[27] website (http://aging.brain-map.org/donors/summary). Information about the primers used can be found in Supplementary Data 10. Cases assessed were a subgroup of those used for the gene expression analyses (control n = 7; AD = 8) because not all the cases have both RNAseq and ISH data available. Subject H14.09.098 had poor staining signal for GAT1 which produced an artifactually large vGluT1/GAT1 ratio, as evidenced by its identification as an outlier by Mahalanobis distances UCL = 2.56 (Supplementary Fig. S6), thus this subject was not included in the analyses. For each case, the vGluT1 and GAT1 images were cropped to the same size sample field (e.g., 9,000,000 µm²) to encompass all layers 1-VI of PCx; the location of each sample field was matched between the images for each gene. Automated counts of cells were performed using scikit-image 0.16.2 and Python 3.7[76]. Images were Gaussian blurred at 3 pixel sigma to remove small imaging artifacts and then thresholded using a Yen intensity threshold[77], which was found to be optimal across numerous thresholding methods blind-tested across several images. Cell objects were counted if they were larger than 10 pixels (equivalent to 10 µm) to identify both small and larger labeled profiles; this threshold size was chosen for final analyses to compensate for potential differences in cell body size between groups, and because a comparison between 10 and 30 µm² size thresholds yielded similar differences in E/I ratios between groups (Supplementary Fig. S5). Values were expressed per 10,000 µm². Nonparametric Wilcoxon test was used for statistical analyses.

**Reporting summary**. Further information on research design is available in the Nature Research Reporting Summary linked to this article.

## Data availability
Source Data are provided with this paper in the "source data file" and Supplementary Data 11. Transcriptomic and ISH data is publicly available from https://aging.brain-map.org. Specifically, normalized RNA seq data can be downloaded from: https://aging.brain-map.org/download/index, and ISH images can be downloaded by accessing case information: from https://aging.brain-map.org/donors/summary. Source data are provided with this paper.

## Code availability
The automated cell counts code is freely available at https://github.com/cdcox/cellCounterForLaturborneta1 under an Apache 2.0 license. The code for quantifying single-labeled immunofluorescence puncta (the "Code") is available upon request to Dr. Lynch; access and use of the Code is subject to a non-exclusive, revocable, non-transferable, and limited right to use for the exclusive purpose of undertaking academic, not-for-profit, or governmental research. Use of the Code or any part thereof for commercial purposes is strictly prohibited in the absence of a Licensing Agreement from The University of California at Irvine.

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

## Acknowledgements

Work was supported by NIA grants AG053740 (to A.L. and J.C.L.) and P50 AG005136 (to C.D.K.), NICHD grant HD079823 (to J.C.L.), NICHD grant HD089491 (to C.M.G. and G.L.), and NIMH grant MH113177 (to A.L.). Additional support was provided by the Weill NeuroHub and the Nancy and Buster Alvord Endowment (to C.D.K.), and the Amon G. Carter Foundation and Peterson-Coutin fund (to A.L.). This work was supported in part by the NIMH Intramural Research Program. The UCI-ADRC is funded by NIH/NIA grant P30 AG066519, and the HBSFRC at the VA West Los Angeles Healthcare Center was supported by NIH/NINDS grant NS056062. We thank the Allen Institute for Brain Science for making all data publicly available.

## Author contributions

J.C.L. and A.L. conceived and designed all studies, and were the primary authors; A.L., J.C.L., P.S., C.D.K., and C.D.C. conducted research; A.S. implemented a priori GO differential gene expression and WGCNA; G.L. and C.M.G. contributed to data interpretation and co-authored the manuscript.

## Competing interests

The authors declare no competing interests.
