## [Peer Review File · Nature Communications]

Reviewer #1 (Remarks to the Author):

Lauterborn et al., describes functional and anatomical synaptic alterations of the excitatory (E) and inhibitory (I) components in post-mortem samples from Alzheimer's disease (AD), Down syndrome (DS) and controls using fluorescence deconvolution tomography and microtransplantation of synaptic membranes into oocytes. The authors found elevated functional and anatomical E/I ratios in AD samples relative to controls. Transcriptional changes in the excitatory and inhibitory components were consistent with the functional increase in E/I ratios and associated with cognitive scores and AD pathology. The authors conclude this is the first evidence of first evidence of pro-excitatory perturbation of synaptic E/I balance in AD parietal cortex and support the hypothesis that E/I imbalances disrupt cognition in AD.

This is a relevant set of data since the manuscript addresses directly the function of the excitatory and inhibitory synaptic components from post-mortem AD and control human samples. Thus, providing relevant electrophysiological data from humans. In addition, the main conclusions are reinforced by anatomical and transcriptome data. The selection of the inferior parietal cortex as a region of study adds significant functional relevance since this region is a hub for AD-associated pathology and default mode network functional alterations. The conclusions are well supported, and the authors provide a balance and detailed discussion of the results.

I have only a few minor suggestions:

- 1) Introduction, "Our previous work showed the total activity of functional GABAARs, both synaptic and extrasynaptic, is significantly reduced in the temporal cortex of the AD brain, and that this reduction is equal to or greater than the reduction in AMPARs [12]." The concept of "equal" or "greater" is not clear since these two terms are exclusive.
- 2) Define better "middle-aged" AD subjects, particularly since this cohort seems to be composed by early-onset AD cases (age <65 years; Table 1). Early-onset AD cases are not "middle-aged AD subjects". Please confirm if these cases are FAD subjects.
- 3) Since there are only 16 cases, the Table 1 should describe each case instead of group averages.
- 4) "normal controls without pathology". The term "normal controls" needs clarification or a better term. For example, "non-demented subjects without AD pathology".
- 5) Controls are described as "normal controls without pathology" but there is no AD pathology or cognitive scores for these "controls". Please clarify if controls refer to non-demented individuals and whether AD pathology was assessed. This is particularly relevant because non-demented individuals may have AD pathology.
- 6) "Stages VI/C", please add information about scales used for the AD pathology and severity (e.g, Braak, CERAD, plaques, tangles). These are global scales, it would be important to describe AD pathology in the studied region. These results could be incorporated in Figure 1.
- 7) Results, Figure 1, "Nevertheless, in all cases there was evidence of age-related pathology including an accumulation of large lipofuscin deposits often in association with GPHN immunopositive (+) cell bodies." There is no data showing age-related effects in Figure 1 that support this conclusion. In addition, since only "middle-aged" are included with same AD pathology (Table 1), how age-related effects were determined?
- 8) Figure 1, define better the x-axis (e.g., ir-intensity?). Please also define high, mid, and low in Figure 2 (e.g. ir-intensity?).
- 9) "(≥ 90 immunofluorescence intensity, on a 20-180 scale)". It would be better to use percentiles rather than 75 and 90 in a 20-180 scale. Figure 2, the cut off <75 or >90 has no real meaning since

the scale is 180. Using low, mid, and high indicating what is being measured (e.g., ir-intensity) would create less confusion.

10) There are Figures with p values and other without. Please add p values in Figure 1 to be consistent and to indicate that those differences are significant.

11) Figure 4e-g, please add "synaptosomes" to small, medium and large to clarify the graph and y-axis.

12) Figure 4g, add what is being measured in the y-axis and layer. High IR / Low IR of what? PSD-95, PGHN, Layer 1, Layer 2? In general, the x- and y-axes are not well defined. For example, Figure 2 uses "High/Low IR" in the y-axis with the info about GPHN/PSD95 and layer in the title, Figure 4 uses a different term "High IR / Low IR" in the y-axis but no info about proteins or layers and no title, and Figure 3 proteins are in the y-axis but no info about "IR" or what is being measured.

13) Figure 4g, add the color key for blue and purple.

14) Trace responses in Figure 5a does not match the quantification in Figure 5b. Please use traces that match the quantifications.

15) Figure 5f, add what is being measured (e.g., Kainate/GABA currents, max response, slope? "Kainate/GABA" does not indicate what is being measured. Similarly, in Figure 5d-e "peak" needs additional information, "current". Is "peak" in 5d and "response" in 5b the same measures? Is "Kai" and "Kainate" similar? Please be consistent and label properly the x- and y-axes.

16) Figure 6a-B, add what is being measured in y-axis (mRNA levels?). Figure 6b, why is there an asterisk for the No Dementia in stage 0 but not in stage 3 or figure 6a? Please be consistent across Figures indicating p values and asterisks.

17) The results for Figure 6c need a better description in the main text. The y-axis labeling cannot be understood. The abbreviation cannot be in a supplementary table, particularly when there is room to add this information.

Reviewer #2 (Remarks to the Author):

The authors have appropriately identified a level of analysis that has been left largely unexplored and undetermined in human brain tissue from Alzheimer's Disease patients.

That is, the question whether there is evidence for an impaired excitatory-inhibitory balance as some clinical phenotypes (e.g., epilepsy in AD patients) and several mouse model studies are suggesting.

The authors fill this void by analysing human tissue (from AD and Down Syndrome patients) from an anatomical, electrophysiological and transcriptomic perspective. The combination of an interesting set of experiments is a strength of the paper.

In its current form, the manuscript also has a number of weaknesses, which diminish my enthusiasm for it.

My primary concern is that key findings (especially fig 3b, fig 5f) rely on low n-numbers and are barely significant,

which leaves me unsure about the robustness and relevance of the presented findings.

ANATOMICAL ANALYSIS

Fig 3b: this is a key primary outcome for the authors, and while it may be statistically significant (but $p=0.0048$) I wonder if the experiment is

robust and reproducible. It would be more convincing if the authors would include other

excitatory/inhibitory markers as positive controls.

ELECTROPHYSIOLOGY

In fig 5f apparently 5 measurements/data points from 5 AD patients were included in the analysis, the data was highly variable, only 3/5 data points were above controls and the p value is barely significant.

So similar to my comment for fig 3b, I would like to get some reassurance that the effect is robust and reproducible.

Also, it is not clear to me how many oocytes were measured per patient, and how many recordings were averaged for the results in fig 5b.

TRANSCRIPTOMICS

To see if findings are true in multiple populations, it would have been important to check more than one dataset (since the authors also use the plural "datasets" in the abstract).

In addition, the authors should/could give a better picture of the overall excitatory vs inhibitory gene expression in AD vs non-AD, for example by looking at co-expressed gene modules varying between the two groups (to go beyond the ratio of PSD95/gephrin).

In Figure 6a, it appears to me that groups were split based on DSM-IV clinical diagnosis, so that authors could compare different levels of amyloid deposition between no dementia (ND) and dementia (D). Looking at the supplementary table, it seems that the DSM-IV classification puts, for example, a person (H14.09.019) with Braak 6 and CERAD 3 into the ND group and another (H14.09.002) with Braak 1 and CERAD 0 in the D group. Can the authors clarify this?

Also, in fig 6a, the authors should not use a t-test to compare ratios of numbers, they should first do a log transformation first. Also, if they want to t-test they should check if the populations are normally distributed.

Reviewer #3 (Remarks to the Author):

Lauterborn et al. investigated excitatory to inhibitory balance in postmortem parietal cortex samples from controls, Alzheimer's disease (AD) cases, and individuals with Down syndrome (DS) and AD neuropathology. The authors found molecular and functional evidence of a pro-excitatory state in AD, but not DS, and associations between E/I imbalance and cognitive decline. The novel aspects of the approach include pairing fluorescence deconvolution tomography (FDT) with microtransplantation of synaptic membranes (MSM) using neighboring samples from the same postmortem tissue, and methods to test electrophysiological changes predicted by the anatomical changes observed. The authors used PSD-95 as an indicator of excitatory postsynaptic densities and gephyrin (GPHN), as an indicator of inhibitory postsynaptic densities. The results are interesting and possibly noteworthy; however there are concerns about the methodology and data presentation that need to be addressed as follows:

1. To be certain that the relative changes in post-synaptic densities in AD are pro-excitatory, the study would need to separate excitatory from inhibitory cells in the analysis. Could additional data be included to support the hypothesis? Given this concern, the claims of being the first to provide evidence of a marked pro-excitatory perturbation of synaptic E/I balance in AD parietal cortex needs to be removed.
2. Assessing ratios of excitatory to inhibitory markers (Fig 2) and electrophysiological changes (Fig 5f)

is unconventional. This is a second level of analysis and exaggerates the differences between groups. The differences in E/I balance are not apparent when looking at the primary data.

3. Disease and control tissue came from different sites which introduces biases. The controls are mostly male veterans and have a much longer post-mortem interval (PMI). It would be important to include additional quality control to provide evidence of the integrity of the tissue with longer PMI.

4. Pearson correlations in Figures 4g, 5d, and 5d were done with all groups combined. The significant correlations appear to be strongly influenced by group differences; this is particularly evident in 4g where the controls cluster in the top right. It would be better to run correlations on each group separately.

5. In the electrophysiological recordings from MSMs, to calculate the E/I ratios, data were collected and averaged only from oocytes in which both kainate and GABA currents were unambiguously measured. Could the authors provide more information about how often unambiguous measures were obtained, and how much data was excluded?

Minor comments are as follows:

1. The discussion should note that the cases were very young which might bias cohorts to have cases with higher likelihood of pro-excitatory E/I imbalance, given the known higher risk of seizures in early-onset AD.

2. Did the controls have neuropathological evaluations? If so, this information should be included.

3. The findings that variability in ion currents correlated with p-tau levels in the DS tissue but not AD needs more explanation. What is the rationale for these comparisons? The raw data plots should be included.

Reviewer #4 (Remarks to the Author):

I have two major concerns on this manuscript.

The first one is related to the ratio of expression level of a gene pair. It is well known (see <https://doi.org/10.15252/msb.20167144>) that the relationship between transcript level and protein concentration is gene-specific and, furthermore, cannot be linearly correlated with the relevant activity for several reasons (e.g. post-translational modifications, protein-protein interactions, etc.). In this specific case the implication for the excitatory / inhibitory relationship of the synapses is not granted. The Authors mention in the methods that "Additional histopathological data and protein quantification from the same subjects was downloaded from the same site" but is not clear if quantified proteins are the same considered in the transcript ratio or are other proteins involved in the disease.

I suggest to verify if the expression level of DLG4 is significantly and consistently higher and conversely that the expression level of GPHN is consistently lower in patients.

The second concern is related to the sample size. Patients are only 28 stratified in 4 categories, and comparisons of gene expression ratios are carried out in each category comparing patients with and without dementia. Apparently these numbers are too small for a robust statistic assessment. It is also not clear why the interred ratio is opposite in the first stage of the disease.

Reviewers' comments:

Reviewer #1 (Remarks to the Author):

Lauterborn et al., describes functional and anatomical synaptic alterations of the excitatory (E) and inhibitory (I) components in post-mortem samples from Alzheimer's disease (AD), Down syndrome (DS) and controls using fluorescence deconvolution tomography and microtransplantation of synaptic membranes into oocytes. The authors found elevated functional and anatomical E/I ratios in AD samples relative to controls. Transcriptional changes in the excitatory and inhibitory components were consistent with the functional increase in E/I ratios and associated with cognitive scores and AD pathology. The authors conclude this is the first evidence of first evidence of pro-excitatory perturbation of synaptic E/I balance in AD parietal cortex and support the hypothesis that E/I imbalances disrupt cognition in AD.

This is a relevant set of data since the manuscript addresses directly the function of the excitatory and inhibitory synaptic components from post-mortem AD and control human samples. Thus, providing relevant electrophysiological data from humans. In addition, the main conclusions are reinforced by anatomical and transcriptome data. The selection of the inferior parietal cortex as a region of study adds significant functional relevance since this region is a hub for AD-associated pathology and default mode network functional alterations. The conclusions are well supported, and the authors provide a balance and detailed discussion of the results.

I have only a few minor suggestions:

1) Introduction, "Our previous work showed the total activity of functional GABA_ARs, both synaptic and extrasynaptic, is significantly reduced in the temporal cortex of the AD brain, and that this reduction is equal to or greater than the reduction in AMPARs [12]." The concept of "equal" or "greater" is not clear since these two terms are exclusive.

Response: We agree and have modified the sentence as follows:

"Our previous work showed the *total* activity of functional GABA_ARs, both synaptic and extrasynaptic, is significantly reduced in the temporal cortex of the AD brain, and that this reduction is similar in magnitude to the reduction in AMPARs"

2) Define better "middle-aged" AD subjects, particularly since this cohort seems to be composed by early-onset AD cases (age <65 years; Table 1). Early-onset AD cases are not "middle-aged AD subjects". Please confirm if these cases are FAD subjects.

Response: We appreciate the reviewer's comment, and the opportunity to clarify the cases more. We used the term "middle-aged" most often in reference to all three groups, as that seemed an appropriate identifier for the collective cases. We agree that the AD group should be defined "early-onset" based on their age and have made that change in the text where appropriate; in some cases, we have retained "middle-aged" to indicate the age of the subjects. Regarding FAD, 3 out of 5 AD cases have been genotyped and are not FAD subjects. We were unable to genotype the last two cases due to the lengthy COVID-19 shut down of UCI and the tissue bank, although they are now in the process of having that done (but we don't have a firm due date because of COVID). However, in reviewing our data it is important to point out that for the kainate/GABA ratios (Fig 5f) the 3 non-FAD cases were fairly distributed (highest, third highest, and lowest) on this measure for the group, as they were on other measures. So, for our cases we can only conclude that the findings are associated with early-onset but not with known FAD.

3) Since there are only 16 cases, the Table 1 should describe each case instead of group averages.

Response: We agree with the reviewer and have included a supplementary table with the description of each case. We believe the summary table provides a fast glance to the cohort directly within the manuscript, while the reader can easily access each individual information in the new added supplementary data 1.

4) *“normal controls without pathology”*. The term *“normal controls”* needs clarification or a better term. For example, *“non-demented subjects without AD pathology”*. and

5) *Controls are described as “normal controls without pathology” but there is no AD pathology or cognitive scores for these “controls”*. Please clarify if controls refer to non-demented individuals and whether AD pathology was assessed. This is particularly relevant because non-demented individuals may have AD pathology

Response to points 4&5: We agree that “normal controls” needed better clarification. Our control tissue came from VA subjects that died primarily from disease (e.g., cancer, breathing disorder). These cases did not have any information in their clinical summaries to indicate cognitive issues and, expectedly, there was no reported cognitive testing. Gross and microscopic evaluations of these brains were conducted by the HBSFRC neuropathologist and, in all cases, there was no evidence of neuropathology, hypoxia or metastatic disease. Therefore, all controls were diagnosed by the pathologist as essentially normal brains. We have added this information to the Methods on the general pathology report, under the “Cases and tissue samples” section.

Additionally, we assessed the levels of amyloid plaques in all cases (new analyses, Fig 1b), and determined that none of the control cases exhibited plaques. Based on this we have defined the controls as subjects without AD pathology (Results, 1st paragraph).

6) *“Stages VI/C”*, please add information about scales used for the AD pathology and severity (e.g, Braak, CERAD, plaques, tangles). These are global scales, it would be important to describe AD pathology in the studied region. These results could be incorporated in Figure 1.

Response: We believe the reviewer is referring to Table 1’s “Stages VI/C”. We have amended the Table 1 description to be more explicit that we are indicating Braak Stage VI and CERAD stage C. Also, as noted above. we have followed the reviewer’s suggestion and have conducted additional analyses for amyloid plaques (Fig 1b). All AD and DS cases exhibited plaques, and there were no group differences. The reader is also referred to the text for further details (Results 1st paragraph).

7) *Results, Figure 1, “Nevertheless, in all cases there was evidence of age-related pathology including an accumulation of large lipofuscin deposits often in association with GPHN immunopositive (+) cell bodies.” There is no data showing age-related effects in Figure 1 that support this conclusion. In addition, since only “middle-aged” are included with same AD pathology (Table 1), how age-related effects were determined?*

Response: Although lipofuscin is considered a reliable marker of brain aging in the field, we agree with the reviewer’s comment. Therefore, we have reworded this section to now state: “Nevertheless, in all cases there were large lipofuscin deposits often in association with GPHN immunopositive (+) cell bodies, and in the AD cases we observed dystrophic GPHN+ processes (not shown).” We thought it important to acknowledge these observations because they were present in the material, although our analyses focused on the synaptic labeling. Because of the addition of the plaque data, we removed the images showing GPHN with lipofuscin/dystrophic processes in the revised Fig 1.

8) *Figure 1, define better the x-axis (e.g., ir-intensity?). Please also define high, mid, and low in Figure 2 (e.g. ir-intensity?).*

Response: The figure caption now states: “(e-h) Synaptic immunolabeling intensity frequency distributions from the FDT analyses show the proportion of PSDs (Y-axis) in layers 1 (e,f) and 2 (g,h) that were immunolabeled for PSD-95 or GPHN at different immunoreactivity (IR) intensities (X-axis)...”. Thus, the “IR

intensity” is a scale that reflects immunofluorescence intensity levels. In addition, we have defined high, mid, and low in Figure 2 as follows: “...with immunofluorescence in low (<75), mid (75-90) or high (>90) intensity ranges for each group (group means \pm SEM; based on Fig. 1e-h IR intensity distributions).”

9) “(≥ 90 immunofluorescence intensity, on a 20-180 scale)”. It would be better to use percentiles rather than 75 and 90 in a 20-180 scale. Figure 2, the cut off <75 or >90 has no real meaning since the scale is 180. Using low, mid, and high indicating what is being measure (e.g., ir-intensity) would create less confusion.

Response: The intensity values are a numerical scale based on the intensity of the immunofluorescence from 8-bit images; the usual scale is 0-256 but our labeling generally falls within the 20-180 range. From this perspective, we believe that using percentiles for the X-axis in figure 1 could be somewhat mis-leading, and also make it more difficult to relate figures 1 and 2. For figure 2, we agree with the reviewer that it would be less confusing to use low, mid, and high alone, and to indicate that these categories reflect the levels of ir-intensity. We have made these changes to the figure.

10) There are Figures with p values and other without. Please add p values in Figure 1 to be consistent and to indicate that those differences are significant.

Response: The RM-ANOVA p-values for the interaction between the groups and intensities for each protein and in each field (panels e-h) were presented in the figure caption, and all were highly significant; we would respectively argue that this is the most important statistical outcome to present for these panels. We did not conduct posthoc tests at all individual intensity bins since our goal was to evaluate the relative shift between the puncta with high IR intensity range levels versus low IR intensity range levels, as shown in Figure 2. But, so as to not mislead the reader, we have removed the arrows in panels g and h that emphasized the controls groups.

11) Figure 4e-g, please add “synaptosomes” to small, medium and large to clarify the graph and y-axis.

Response: We have added this information to the y axis to the figure to be consistent.

12) Figure 4g, add what is being measured in the y-axis and layer. High IR / Low IR of what? PSD-95, GPHN, Layer 1, Layer 2? In general, the x- and y-axes are not well defined. For example, Figure 2 uses “High/Low IR” in the y-axis with the info about GPHN/PSD95 and layer in the title, Figure 4 uses a different term “High IR / Low IR” in the y-axis but no info about proteins or layers and no title, and Figure 3 proteins are in the y-axis but no info about “IR” or what is being measured.

Response: Figure 4g caption defined the y- axis “High IR / Low IR” as “FDT high-to-low immunoreactivity (IR) ratios (PSD-95-ir and GPHN-ir in layers 1 and 2 combined)”. We have modified the y-axis label to read “High/Low IR (PSD-95 + GPHN)” and modified the figure caption to better clarify that we grouped all synaptic FDT data for both proteins and both fields. The “High/Low IR” is now consistent with Figure 2. As described in Figure caption 3, “the y-axis represents the ratio of the peak value from the PSD-95 intensity frequency distribution to the peak value from the GPHN distribution”. We have expanded this sentence to clarify what is being shown and have changed the y-axis to read “Peak PSD-95+ / GPHN+ Synapse Counts”. Since we defined IR as immunoreactivity, we are using “+” to represent “immunopositive” because we are showing the ratio of peak numbers of PSD-95+ synapses to peak numbers of GPHN+ synapses derived from the intensity frequency histograms.

13) Figure 4g, add the color key for blue and purple.

Response: It has been added for all three groups.

14) Trace responses in Figure 5a does not match the quantification in Figure 5b. Please use traces that match the quantifications.

Response: We are grateful to the reviewer for catching this mismatch. There was an error in the scale of the plot that has been corrected. The currents in the traces for the AD group are shown at different scale to appreciate better the currents.

15) Figure 5f, add what is being measured (e.g., Kainate/GABA currents, max response, slope? “Kainate/GABA” does not indicate what is being measured. Similarly, in Figure 5d-e “peak” needs additional information, “current”. Is “peak” in 5d and “response” in 5b the same measures? Is “Kai” and “Kainate” similar? Please be consistent and label properly the x- and y-axes.

Response: We have modified Fig 5F y-axis to read “Kai/GABA Maximum Amplitudes”, and in figs 5d-e we have relabeled the x- and y-axis to indicate that they are showing the maximum amplitude responses. We have defined “Kai” as kainate in the figure caption.

16) Figure 6a-B, add what is being measured in y-axis (mRNA levels?). Figure 6b, why is there an asterisk for the No Dementia in stage 0 but not in stage 3 or figure 6a? Please be consistent across Figures indicating p values and asterisks.

and 17) The results for Figure 6c need a better description in the main text. The y-axis labeling cannot be understood. The abbreviation cannot be in a supplementary table, particularly when there is room to add this information.

Response to points 16&17: Following the advice of Reviewers 2, 3 and 4 we have reanalyzed all transcriptomic data at three different levels. The revised Fig 6 shows the new analysis, which now only focuses on differences between controls with no pathological change and AD cases, and all efforts were done to preserve consistency in displaying the statistical results across figures. The revised Results section “Increased transcriptomic DLG4/GPHN ratio in AD” (beginning at bottom of page 8) has been rewritten to incorporate these new analyses and additional information as requested.

Reviewer #2's comments:

The authors have appropriately identified a level of analysis that has been left largely unexplored and undetermined in human brain tissue from Alzheimer's Disease patients.

That is, the question whether there is evidence for an impaired excitatory-inhibitory balance as some clinical phenotypes (e.g., epilepsy in AD patients) and several mouse model studies are suggesting.

The authors fill this void by analysing human tissue (from AD and Down Syndrome patients) from an anatomical, electrophysiological and transcriptomic perspective. The combination of an interesting set of experiments is a strength of the paper.

In its current form, the manuscript also has a number of weaknesses, which diminish my enthusiasm for it. My primary concern is that key findings (especially fig 3b, fig 5f) rely on low n-numbers and are barely significant, which leaves me unsure about the robustness and relevance of the presented findings.

1. ANATOMICAL ANALYSIS

Fig 3b: this is a key primary outcome for the authors, and while it may be statistically significant (but $p=0.0048$) I wonder if the experiment is robust and reproducible. It would be more convincing if the authors would include other excitatory/inhibitory markers as positive controls.

Response: The primary reason we focused on PSD-95 and GPHN was that these proteins are integral to excitatory and inhibitory synapses, respectively, regardless of the presence of other markers such as excitatory and inhibitory receptors that might vary across synaptic pools such as cortical layers. But, at this point we are

unable to measure additional proteins from the same tissue blocks that were used for the PSD-95 and GPHN analyses due to insufficient tissue for all groups. However, we appreciate the reviewer's point as to the robustness and reproducibility of our key findings. To address this, we conducted new E/I analyses on in situ hybridization material for the expression of the vesicular glutamate transporter vGluT1 and the GABA transporter GAT1 in parietal cortex in control and AD cases (n = 8 controls and 12 AD); for these studies we used the publicly available ADTBI study cohort material. As shown in new Figure 7 and Supplementary Figure 4, and described in the Results (page 10) ratios of vGluT1 to GAT1 expressing cells in parietal cortex demonstrated a significant increase in this E/I measure in AD versus the control group. Interesting, this effect was mainly driven by a reduction in GAT1 expressing cells but consistent with our FDT findings (Fig 3). These new results support the conclusion that the synaptic changes in PSD95 and GPHN in AD are among more widespread cellular changes including marked decreases in GAT1 expression that contribute to the shift in E/I balance.

We believe that these independent analyses, coming from a separate larger cohort and assessing excitatory and inhibitory cells themselves, provide strong corroborating evidence for the FDT and MSM findings.

2. ELECTROPHYSIOLOGY

In fig 5f apparently 5 measurements/data points from 5 AD patients were included in the analysis, the data was highly variable, only 3/5 data points were above controls and the p value is barely significant...I would like to get some reassurance that the effect is robust and reproducible. Also, it is not clear to me how many oocytes were measured per patient, and how many recordings were averaged for the results in fig 5b.

Response: We originally used the average of three oocytes per group, and each oocyte was measured at least twice for each current when Dr Limon was at UC Irvine. When he moved to UTMB, the experiments were repeated with more oocytes per subject, and the results were highly reproducible, so results from UCI and UTMB were combined. The experiments were done blind by different researchers with the results consistent across institutions. That being said the electrophysiological E/I results were significant for the AD versus control group comparison and consistent with the FDT studies. The number of oocytes is now listed in supplementary data 2).

3. TRANSCRIPTOMICS

To see if findings are true in multiple populations, it would have been important to check more than one dataset (since the authors also use the plural "datasets" in the abstract).

Response: We apologize for the confusion. We originally used the term "datasets" because each subject has a single dataset; we have corrected the abstract in the revised manuscript. It is also important to note that we did look for additional datasets in GO and we could not find additional studies that have RNA Sequencing of the parietal cortex with enough subjects for robust analyses. Most of the available studies focus on the temporal cortex and the hippocampus that are out of the scope of this paper. However, we have extended our analyses to also include vGluT1 and GAT1 in situ hybridization which also demonstrated significantly elevated E/I for AD vs control groups.

In addition, the authors should/could give a better picture of the overall excitatory vs inhibitory gene expression in AD vs non-AD, for example by looking at co-expressed gene modules varying between the two groups (to go beyond the ratio of PSD95/gephrin).

Response: We thank the reviewer for this suggestion and have included a comprehensive transcriptomic analysis using three different strategies. 1) multivariate correlation analysis of the transcriptome in the parietal cortex with the DLG4/GPHN ratio followed by gene ontology (GO) of top correlated genes to determine category representation, 2) fold change of gene expression in predetermined excitatory and inhibitory GO modules and 3) weighted gene co-expression analysis (WCGNA) to determine unbiased modules of gene co-

expression. We also included in situ hybridization to count excitatory and inhibitory cells in adjacent sections of those used for RNAseq experiments. All these analyses converge to show reduced gephyrin expression, alterations in postsynaptic densities, and cellular changes correlated with loss of inhibitory neurons.

In Figure 6a, it appears to me that groups were split based on DSM-IV clinical diagnosis, so that authors could compare different levels of amyloid deposition between no dementia (ND) and dementia (D). Looking at the supplementary table, it seems that the DSM-IV classification puts, for example, a person (H14.09.019) with Braak 6 and CERAD 3 into the ND group and another (H14.09.002) with Braak 1 and CERAD 0 in the D group. Can the authors clarify this?

Response: We apologize for not clarifying better in our original manuscript. The ADTBI cohort contains cases representative of clinical settings. For example, case H14.09.019 belongs to the category of Non-Demented with AD pathology (NDAN) subjects or resilient (Bjorklund, et al 2012. Mol Neurodegener 7: p. 23). People with clinical diagnosis of AD but not AD pathology also are present in this cohort (Nelson et al 2011. Acta Neuropathol, 121:571-587). In the revised manuscript, and attending to the comments of reviewer 4, we use a subset of cases that better reflect the characteristics of the cohort used in FDT and MSM experiment and thus limited our analysis to control (CERAD 0, no dementia) vs AD cases (CERAD 3 and dementia). This allows for a more direct comparison between the first cohort in FDT and MSM experiments and the second cohort in mRNA expression.

Also, in fig 6a, the authors should not use a t-test to compare ratios of numbers, they should first do a log transformation first. Also, if they want to t-test they should check if the populations are normally distributed.

Response: In the revised analyses we used non-parametric tests for the comparisons of the E/I ratio as well as for the GPHN and DLG4 expression levels. Then we performed the differential expression and WGCNA using transformed data.

Reviewer #3's comments:

Lauterborn et al. investigated excitatory to inhibitory balance in postmortem parietal cortex samples from controls, Alzheimer's disease (AD) cases, and individuals with Down syndrome (DS) and AD neuropathology. The authors found molecular and functional evidence of a pro-excitatory state in AD, but not DS, and associations between E/I imbalance and cognitive decline. The novel aspects of the approach include pairing fluorescence deconvolution tomography (FDT) with microtransplantation of synaptic membranes (MSM) using neighboring samples from the same postmortem tissue, and methods to test electrophysiological changes predicted by the anatomical changes observed. The authors used PSD-95 as an indicator of excitatory postsynaptic densities and gephyrin (GPHN), as an indicator of inhibitory postsynaptic densities. The results are interesting and possibly noteworthy; however there are concerns about the methodology and data presentation that need to be addressed as follows:

1. To be certain that the relative changes in post-synaptic densities in AD are pro-excitatory, the study would need to separate excitatory from inhibitory cells in the analysis. Could additional data be included to support the hypothesis? Given this concern, the claims of being the first to provide evidence of a marked pro-excitatory perturbation of synaptic E/I balance in AD parietal cortex needs to be removed.

Response: We thank Reviewer 3 for the very helpful suggestion to assess excitatory and inhibitory cells, which overlapped Reviewer 2's suggestion for additional analyses of excitatory and inhibitory markers. As described above, we did that by evaluating vGluT1 and GAT1 expressing cells in parietal cortex. We believe that with the combination of results, including the synaptic measures (FDT, MSM) and the cell body measures (gene expression) for excitatory and inhibitory markers, our findings do provide the first evidence of a marked pro-excitatory perturbation of synaptic E/I balance in AD parietal cortex.

2. Assessing ratios of excitatory to inhibitory markers (Fig 2) and electrophysiological changes (Fig 5f) is unconventional. This is a second level of analysis and exaggerates the differences between groups. The differences in E/I balance are not apparent when looking at the primary data.

Response: In presenting the results, we believed it important to show the group data for each synaptic measure because both were altered, although the main point of the analyses was to investigate the E/I balance that can only be done by deriving the ratios for each subject. Figure 2 shows group data for PSD95+ and GPHN+ synapses, with high levels of each synaptic marker being reduced in the AD and DS groups (this is best demonstrated in the high to low IR comparison for each marker; Fig 2b-h). Because both proteins levels were affected, and both were measured in the same field and 3D builds, we believe that assessing them relative to each other is appropriate for each individual case. These resultant E/I findings for PSD95/GPHN by case are shown in Figure 3. These sequential analyses demonstrated a surprising point that while levels of PSD95 and GPHN were both reduced at AD synapses, the PSD95/GPHN ratios of these two synaptic markers, in the same sample field, were elevated in the AD group. Synapse loss is a feature of AD so reductions in synaptic markers are not surprising. But this is the first evidence we are aware of showing the shift in synaptic E/I balance.

3. Disease and control tissue came from different sites which introduces biases. The controls are mostly male veterans and have a much longer post-mortem interval (PMI). It would be important to include additional quality control to provide evidence of the integrity of the tissue with longer PMI.

Response: We appreciate the Reviewer's comment. Sourcing control tissue from another bank was necessary because the UCI ADRC bank doesn't have control tissue for the age range we examined; DS tissue is also exceptionally difficult to get from other banks and thus we could not source all tissues from one bank. Although the controls had longer PMIs, it is important to point out that the controls exhibited greater numbers of synaptosomes (Fig. 4), and greater numbers of high-to-low intensity PSD-95-ir and GPHN-ir synapses (Figs. 1 & 2) relative to the AD & DS cases indicating preservation of protein content. Moreover, in a recent study that utilized these same controls samples as part of a larger group we found no evidence for effects of PMI (<20 hrs) alone on several synaptic proteins (Lauterborn et al., 2020, Brain Pathology 30, 319-331); this is cited in the current manuscript in the Methods (page 14). As to other measures of tissue integrity, the controls had RIN values between 6.6-8.6 (median = 7.9), indicating relatively good tissue quality; we have now included this in the Methods (page 14). Finally, in a recently published paper by our group (Scaduto et al., 2020, Scientific Reports, 10, p8626) we found no effect of PMI on kainate and GABA responses in human MSM experiments through 87hrs; also referenced in the Methods (page 14). Collectively, we would argue that the longer PMI in the controls had no influence on our main observation that E/I balance in the AD group was elevated (due to protein losses).

4. Pearson correlations in Figures 4g, 5d, and 5d were done with all groups combined. The significant correlations appear to be strongly influenced by group differences; this is particularly evident in 4g where the controls cluster in the top right. It would be better to run correlations on each group separately.

Response: We agree with the reviewer that the correlations are driven by differences between groups which provide visual support for our results. Unfortunately, the limited n in each group is not enough to perform correlations with reliable statistical power.

5. In the electrophysiological recordings from MSMs, to calculate the E/I ratios, data were collected and averaged only from oocytes in which both kainate and GABA currents were unambiguously measured. Could the authors provide more information about how often unambiguous measures were obtained, and how much data was excluded?

Response: In our system we have typically large responses elicited by the application of GABA and smaller responses elicited by kainate. Due to the strong correlation between GABA and kainate currents in oocytes and

samples, few oocytes with small GABA currents had kainate currents below the detection limit (<0.3 nA) of our electrophysiological system and thus unusable to calculate the E/I ratio. In oocytes with detectable but small kainate currents, only ion currents with a clear activation and deactivation phases, a signal to noise ratio of at least 3:1 and, consistently activated by multiple applications in the same oocyte were used for the E/I estimation. We aimed to measure the E/I ratio in at least 4 oocytes per subject. No recorded oocyte was excluded. If the oocyte had GABA but not detectable Kainate response, the GABA response was included in the GABA current measurements. That explains the larger number of oocytes for GABA responses in supplementary data 2. The total number of oocytes for each measurement is now included in supplementary table 2. We have included this information in page 17 of the text (Methods).

Minor comments are as follows:

1. The discussion should note that the cases were very young which might bias cohorts to have cases with higher likelihood of pro-excitatory E/I imbalance, given the known higher risk of seizures in early-onset AD.

Response: This is an excellent point that we have now incorporated into the Discussion (1st paragraph) as follows: “Notably, while we cannot exclude the possibility that these synaptic E/I effects might reflect a feature of early-onset AD such as seizure activity, complementary findings from the ADTBI cohort suggest that E/I imbalance is not an exclusive feature of early-onset AD”.

Given that the rare autosomal dominant forms of AD have a significant risk for developing seizures, but that the 3 AD cases tested were not FAD, it is likely that our findings for the early-onset AD group were not a reflection of seizure activity. In fact, we would propose to the Reviewer that it is more likely that AD-associated seizures may reflect an elevated E/I balance.

2. Did the controls have neuropathological evaluations? If so, this information should be included.

Response: As stated above (to Reviewer 1), the control brains were examined by a neuropathologist at the HBSFRC. Gross examinations of all control cases indicated no cortical atrophy, normal ventricle size, and no atherosclerosis of vasculature among other non-findings (e.g., no mass, lesions, etc). Microscopic evaluations of H&E stained cortical and hippocampal tissue for all cases found normal neuronal cellularity, with no evidence for neuropathology, hypoxia, or metastatic disease; the neuropathological diagnosis for the controls was that they were essentially normal brains. We have included a statement regarding the neuropathological evaluation in the Methods (2nd paragraph). Additionally, the controls did not exhibit amyloid plaques in the sample field.

3. The findings that variability in ion currents correlated with p-tau levels in the DS tissue but not AD needs more explanation. What is the rationale for these comparisons? The raw data plots should be included.

Response: We have now included the raw data for these comparisons in supplementary data 2. The reason we looked at these two measures relative to each other was because of the variability in ion currents, particularly for the DS group. Previously we had reported that the levels of p-Tau were also more variable in parietal cortex samples from the same DS group, including very high levels, relative to AD and control samples (Lauterborn et al., 2020, Brain Pathology 30, 319-331). So we looked to see if the measures correlated. This proved to be the case: the amplitude of the kainate currents was highly negatively correlated with p-Tau in the DS group, but not in the AD group. These findings not only distinguish the DS and AD groups, but support the conclusion that abnormal hyperphosphorylated tau is associated with negative effects on glutamate receptor function consistent with reports that hyperphosphorylated tau disrupts AMPA receptor trafficking (see Jurado et al., 2017 Front Mol Neurosci. doi: [10.3389/fnmol.2017.00446](https://doi.org/10.3389/fnmol.2017.00446)).

Reviewer #4' comments:

I have two major concerns on this manuscript.

1. The first one is related to the ratio of expression level of a gene pair. It is well known (see <https://doi.org/10.15252/msb.20167144>) that the relationship between transcript level and protein concentration is gene-specific and, furthermore, cannot be linearly correlated with the relevant activity for several reasons (e.g. post-translational modifications, protein-protein interactions, etc.). In this specific case the implication for the excitatory / inhibitory relationship of the synapses is not granted. The Authors mention in the methods that "Additional histopathological data and protein quantification from the same subjects was downloaded from the same site" but is not clear if quantified proteins are the same considered in the transcript ratio or are other proteins involved in the disease. I suggest to verify if the expression level of DLG4 is significantly and consistently higher and conversely that the expression level of GPHN is consistently lower in patients.

Response: We thank the reviewer for these very helpful comments and have included this important point in the manuscript. First, we confirm that mRNA levels for GPHN, but not DLG4, were consistently downregulated in AD, driving the DLG4/GPHN ratio higher in AD (end of first paragraph pg. 9). Then we also used three additional analyses to determine whether alterations in the ratio represent synaptic alterations: 1) multivariate correlation analysis of the transcriptome in the parietal cortex with the DLG4/GPHN ratio followed by gene ontology (GO) of top correlated genes to determine category representation, 2) fold change of gene expression in predetermined excitatory and inhibitory GO modules and 3) weighted gene co-expression analysis (WCGNA) to determine unbiased modules of gene co-expression (pg. 9-10). We also included in situ hybridization to count excitatory and inhibitory cells in adjacent sections of those used for RNAseq experiments (pg. 10). All these analyses converge to show reduced gephyrin expression, alterations in postsynaptic densities, and cellular changes correlated with loss of inhibitory neurons.

2. The second concern is related to the sample size. Patients are only 28 stratified in 4 categories, and comparisons of gene expression ratios are carried out in each category comparing patients with and without dementia. Apparently, these numbers are too small for a robust statistic assessment. It is also not clear why the interred ratio is opposite in the first stage of the disease.

Response: We thank the reviewer for this observation. We have limited our new analyses only to compare the controls with no AD pathology vs demented AD cases (n = 8 controls; 12 AD) which have similar characteristics to the cases that we used for FDT and MSM experiments.

Reviewer #1 (Remarks to the Author):

My comments/suggestions have been properly addressed. I have no additional comments.

Reviewer #2 (Remarks to the Author):

I thank the authors for addressing my comments. The only thing I still do not understand is why they only use two transcripts to derive an E/I ratio, when they could have investigated this aspect by looking at multiple genes (maybe calculating signatures) since they have data on all transcripts. They may have done so to match their initial focus on these two proteins, but they missed the opportunity of providing stronger support and insight to their hypothesis.

Reviewer #4 (Remarks to the Author):

The Authors partially addressed my remarks on the expression level of GPHN and DLG4. Indeed, a validation at protein level is still missing.

Concerning the concern on the sample size, it is of course limited, but the results are significant.

Reviewer #1 (Remarks to the Author):

My comments/suggestions have been properly addressed. I have no additional comments.

We are thankful for the reviewer's previous comments that greatly helped to improve this work.

Reviewer #2 (Remarks to the Author):

I thank the authors for addressing my comments. The only thing I still do not understand is why they only use two transcripts to derive an E/I ratio, when they could have investigated this aspect by looking at multiple genes (maybe calculating signatures) since they have data on all transcripts. They may have done so to match their initial focus on these two proteins, but they missed the opportunity of providing stronger support and insight to their hypothesis.

We are thankful for the reviewer's comments. While the determination of transcriptomic signatures of excitation and inhibition at the synaptic level is a major goal sought after in the field, transcriptomic data of whole tissue still does not have the resolution to address this issue. Many synaptic markers that could potentially be used are differentially expressed across cell types and this variability may confound the analysis. Therefore, as the reviewer noted, we limited the derivation of the transcriptional E/I ratio to PSD-95 and GPHN to match our anatomical analysis in the first cohort. The overlap between the protein detected anatomically by FDT and the transcripts for the same proteins in a distinct cohort provides 1) cross-validation of data between the two cohorts, and 2) maintains the interpretation of our results within the context of the two most abundant and consistently expressed excitatory and inhibitory postsynaptic densities described in the literature. Indeed, this ratio strongly correlates with transcripts for proteins in postsynaptic densities (Fig. 6b). We did use two other "omic" approaches to analyze excitatory and inhibitory modules following the initial reviewer's suggestions. These analyses provide more information about the distinct levels of dysfunction in AD and confirmed the role of GPHN in the synaptic imbalance. We have briefly expanded the methods (Page 19, highlighted in red) to include this information in the revised manuscript.

Reviewer #4 (Remarks to the Author):

The Authors partially addressed my remarks on the expression level of GPHN and DLG4. Indeed, a validation at protein level is still missing. Concerning the concern on the sample size, it is of course limited, but the results are significant.

We are thankful for the reviewer's comments. Our transcriptional analysis was based on

publicly available datasets from the Allen Institute, and there are no protein measures for PSD-95 and gephyrin in the same cohort. The only proteins available in the dataset are toxic peptides like A β and tau and inflammation markers, therefore we cannot do the validation of protein in the same subjects used for the transcriptomic analysis. However, we did measure the expression of PSD-95 and gephyrin by immunohistochemistry and FDT in our first cohort, cross-validating the increase in the PSD-95/GPHN ratio at the protein level specifically in synapses by FDT.

As future studies increase the number of subjects wherein the transcriptome for the parietal cortex and other brain regions part of the default mode network become available, a meta-analysis will be possible to increase the samples size and the robustness of the analyses.